# Non-line-of-sight imaging with arbitrary illumination and detection pattern

Xintong Liu [1], Jianyu Wang [1], Leping Xiao[2,3], Zuoqiang Shi[1,4], Xing Fu [2,3] ✉ & Lingyun Qiu [1,4] ✉

Non-line-of-sight (NLOS) imaging aims at reconstructing targets obscured from the direct line of sight. Existing NLOS imaging algorithms require dense measurements at regular grid points in a large area of the relay surface, which severely hinders their availability to variable relay scenarios in practical applications such as robotic vision, autonomous driving, rescue operations and remote sensing. In this work, we propose a Bayesian framework for NLOS imaging without specific requirements on the spatial pattern of illumination and detection points. By introducing virtual confocal signals, we design a confocal complemented signal-object collaborative regularization (CC-SOCR) algorithm for high-quality reconstructions. Our approach is capable of reconstructing both the albedo and surface normal of the hidden objects with fine details under general relay settings. Moreover, with a regular relay surface, coarse rather than dense measurements are enough for our approach such that the acquisition time can be reduced significantly. As demonstrated in multiple experiments, the proposed framework substantially extends the application range of NLOS imaging.

The technique of imaging objects out of the direct line of sight has attracted increasing attention in recent years[1–26]. A typical non-line-of-sight (NLOS) imaging scenario is looking around the corner with a relay surface, where the target is obscured from the vision of the observer. NLOS imaging aims to recover the albedo and surface normal of the hidden targets with the measured photon information. Potential applications of NLOS imaging include but are not limited to robotic vision, autonomous driving, rescue operations, remote sensing and medical imaging.

To achieve NLOS reconstruction, laser pulses of high temporal resolution are used to illuminate several points on the relay surface, where the first diffuse reflection occurs. After that, photons enter the NLOS domain and are bounced back to the visible surface again by the unknown targets. The hidden targets can be reconstructed with the time-resolved photon intensity measured at several detection points on the visible surface. Commonly used time-resolved detectors are single-photon avalanche diodes (SPAD)[27]. The imaging system is confocal if the illumination point coincides with the detection point for each spatial measurement, otherwise being non-confocal. Besides, we call the measurements regular if the illumination and detection points are uniformly distributed in a rectangular region.

According to the representation of the hidden surface, existing imaging algorithms are divided into three categories: point-cloud-based[28], mesh-based[29] and voxel-based methods[1,8,9,30–35]. Among these categories, voxel-based algorithms yield to be the most efficient ones with low time complexity[32] and fine reconstruction results[34]. For voxel-based methods, the reconstruction domain is discretized with three-dimensional grid points and the albedo is represented as a grid function.

The first voxel-based NLOS reconstruction method is the back-projection algorithm proposed by Velten et al.[1]. The measured

[1]Yau Mathematical Sciences Center, Tsinghua University, Beijing 100084, PR China. [2]State Key Laboratory of Precision Measurement Technology and Instruments, Department of Precision Instrument, Tsinghua University, Beijing 100084, PR China. [3]Key Laboratory of Photonic Control Technology (Tsinghua University), Ministry of Education, Beijing 100084, PR China. [4]Yanqi Lake Beijing Institute of Mathematical Sciences and Applications, Beijing 101408, PR China. ✉e-mail: fuxing@tsinghua.edu.cn; lyqiu@tsinghua.edu.cn

photon intensity is modeled as a linear operator applied to the albedo, and the targets are reconstructed by applying the adjoint operator to the measured data. Further improvements of the back-projection method include rendering approaches for fast implementations[2,16] and filtering techniques[33,36] for noise reduction. The light-cone-transform[30] (LCT) proposed by O'Toole et al. describes the physical process as a convolution of the light cone kernel and the hidden target. In this way, the reconstruction is formulated as a deblurring problem and can be computed efficiently using the fast Fourier transform. The directional light cone transform[31] (D-LCT) generalizes this method and simultaneously reconstructs the albedo and surface normal of the hidden target. The frequency-wavenumber migration[8] (F-K) method uses the wave equation to reconstruct the albedo and can also be implemented efficiently in the frequency domain. The LCT, D-LCT and F-K methods only work directly under confocal settings. Although it is possible to transfer the data collected in non-confocal setups to confocal ones, the approximation error cannot be neglected[34]. To reconstruct the hidden object under non-confocal settings, the phasor field[32] (PF) method formulates the NLOS detection process as one of diffractive wave propagation and provides a direct solution with low time complexity. Its recent extension with SPAD arrays reconstructs live low-latency videos of NLOS scenes[37]. The signal-object collaborative regularization[34] (SOCR) method considers priors on both the reconstructed target and the measured signal, which leads to high-quality reconstruction with little background noise. For scenarios with non-planar relay surfaces, the F-K and back-projection type methods can be used directly. Algorithms designed only for planar relay settings can be applied using the signal shifting techniques[8,14].

Despite these breakthroughs, two major obstacles of existing methods toward practical applications are the need for a large relay surface and dense measurement. If the relay surface is irregular or small, these algorithms may fail due to the lack of data. Besides, dense measurement results in a long acquisition time, which poses a significant challenge for applications such as auto-driving where the observer may move at high speed. In recent works, it was reported that sparse measurements could be used to reconstruct the hidden scenes. Isogawa et al. showed that the target could be reconstructed with confocal and circular NLOS scans[38]. Sparse measurements from square grids scanning on the relay surface could also be used by incorporating the compressed sensing technique[35]. Besides, a single shot can be used to track a moving hidden target[17], although the reconstruction fails when the target is still due to the ill-posedness of the inverse problem.

In this work, we propose a Bayesian framework for NLOS reconstruction which is applicable for any spatial pattern of the illumination and detection points. By introducing the virtual confocal signal at rectangular grid points, we design joint regularizations for the measured signal, virtual confocal signal and the hidden target. We put forward a confocal complemented signal-object collaborative regularization (CC-SOCR) framework, which reconstructs both the albedo and surface normal of the hidden target. The proposed method allows regular and irregular measurement patterns in both confocal and non-confocal scenarios. Besides, our approach provides faithful reconstructions with negligible background noise, even in cases with very coarse and noisy measurements. Notably, the proposed method suggests a paradigm shift, liberating the research of NLOS imaging from relying heavily on the assumption of a large-size relay surface with the entire region (wall, ground). Our method demonstrates high-quality NLOS reconstructions in various scenarios with the relay surfaces having discrete scattering regions, arbitrary irregular shape, or very limited size, enabling the hidden object reconstruction with far more types

of realistic relay surfaces such as window shutter, window frame, and fence, which significantly broadens the scope of NLOS imaging applications. As shown in Fig. 1, the illumination and detection patterns are irregular but manifest in ubiquitous scenes of daily lives. Reconstruction results of the bunny with synthetic confocal signals[39], detected at the entire relay surface and these four scenarios, are provided in Supplementary Figs. 1–5.

## Results

### The NLOS physical model

The goal of NLOS imaging is to take a collection of measured transient data and find the target that comes closest to fitting these measured signals. In this work, we adopt the physical model proposed in SOCR[34]. Let $x_i'$ and $x_d'$ be the illumination and detection points on the visible surface, and we call $(x_i', x_d')$ an active measurement pair, or simply a pair in the following. The photon intensity measured at time $t$ is given by

$$\tau(x_i', x_d', t) = \int_{\Omega} \frac{(x_d' - x) \cdot \mathbf{n}(x)}{|x_i' - x|^2 |x_d' - x|^3} f(x) \delta(|x_i' - x| + |x_d' - x| - ct) dx \quad (1)$$

in which $\Omega$ is the three-dimensional reconstruction domain, $f(x)$ denotes the albedo value of the point $x$, $\mathbf{n}(x)$ is the unit surface normal at $x$ that points towards the visible surface. The unit vector $\mathbf{n}(x)$ can be arbitrarily chosen for points with zero albedo value. By denoting $\mathbf{u} = f\mathbf{n}$, Eq. (1) is written equivalently as

$$\tau(x_i', x_d', t) = \int_{\Omega} \frac{(x_d' - x) \cdot \mathbf{u}(x)}{|x_i' - x|^2 |x_d' - x|^3} \delta(|x_i' - x| + |x_d' - x| - ct) dx \quad (2)$$

Noting that the intensity is linear with $\mathbf{u}$, the physical model can be written as $\boldsymbol{\tau} = A\mathbf{u}$ in the discrete form. The albedo and surface normal can be obtained directly from $\mathbf{u}$. Indeed, the albedo of a voxel $x$ is given by the norm of the vector $\mathbf{u}(x)$. The surface normal of a voxel $x$ is obtained by normalizing the vector $\mathbf{u}(x)$. The surface normal is not defined where the albedo is zero.

### The measured signal

To reconstruct the hidden target, we consider a collection of $M$ measurements. Let $p_m = (x_m^p, y_m^p, z_m^p)$ be the coordinates of the $m^{th}$ illumination point, in which $x_m^p$, $y_m^p$ and $z_m^p$ are the coordinates in the horizontal, vertical and depth directions. We denote by $q_m = (x_m^q, y_m^q, z_m^q)$ the coordinates of the $m^{th}$ detection point, and call $(p_m, q_m)$ a measurement pair. For each pair, the photon counts of the first $T$ time bins are collected. The coordinates of all measurement pairs are written as $C_{meas} = \{(x_m^p, y_m^p, z_m^p, x_m^q, y_m^q, z_m^q) | m \in [M]\}$, in which we denote by $[M]$ the set $\{1, 2, \ldots, M\}$. Let $\bar{\mathbf{b}}$ be the noisy signal measured at $C_{meas}$. In practice, various types of noise inevitably corrupt the measured signals and significantly degrade the quality of the reconstruction. To mitigate the effects of noise and improve the reconstruction quality, we introduce the estimated signal $\mathbf{b}$ as an approximation of the ideal signal considered at the measured locations. The variable $\mathbf{b}$ is treated as a random vector so that it can be determined under the Bayesian framework. Besides, we denote the simulated signal considered at the set $C_{meas}$ as $A_{\mathbf{b}}\mathbf{u}$, in which $A_{\mathbf{b}}$ is the discrete physical model defined in Eq. (2).

### The virtual confocal signal

We discretize the reconstruction domain $\Omega$ with $V = \{(x_i, y_j, z_k) | i \in [I], j \in [J], k \in [K]\}$, in which $x_i$, $y_j$ and $z_k$ are coordinates of the voxel in the horizontal, vertical and depth directions, respectively. When the number of measurement pairs is small, the solution to the least-squares reconstruction problem may not be unique due to the lack of data. To overcome the rank deficiency of the measurement matrix, we

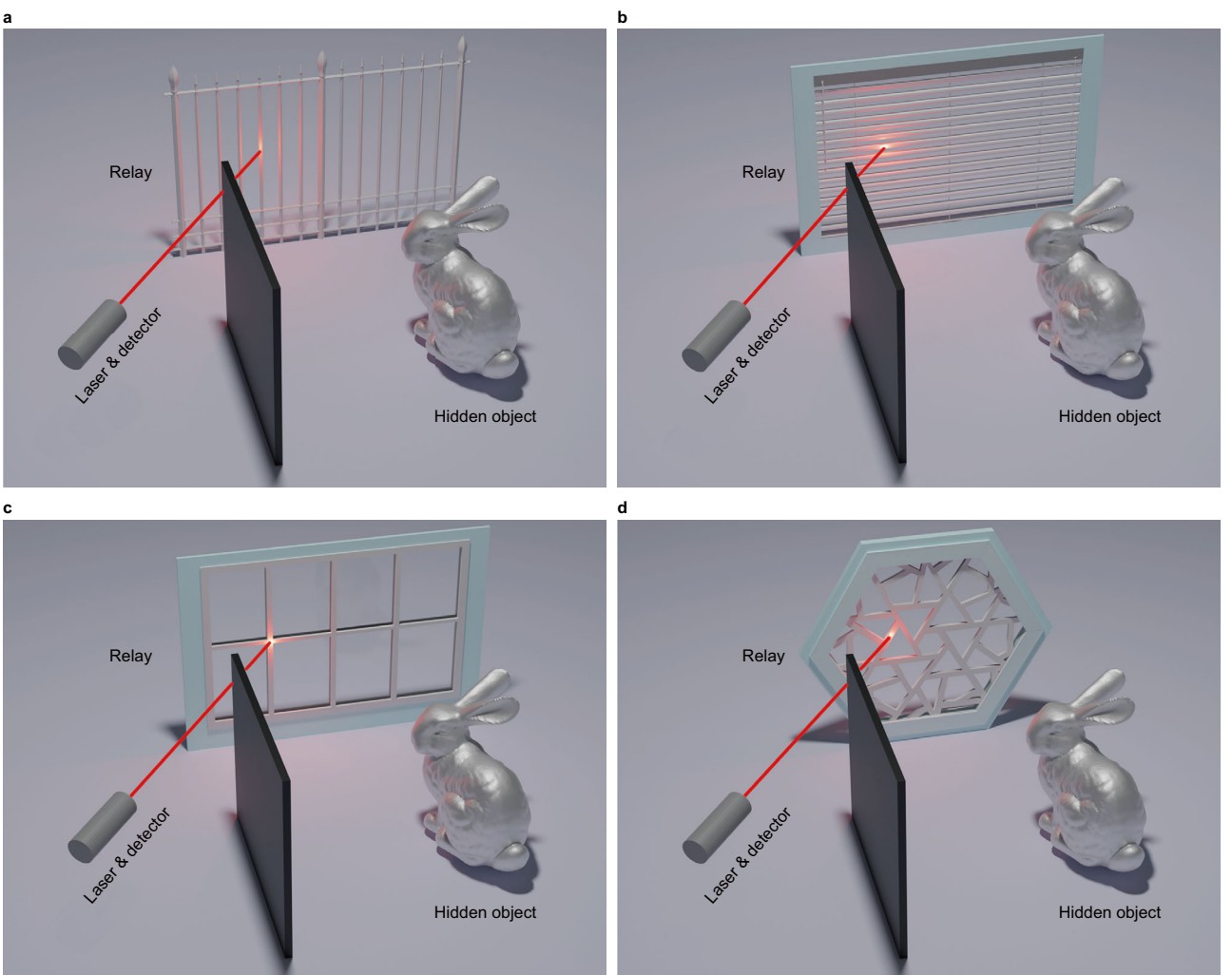

**Fig. 1 | Irregular illumination and detection patterns for NLOS imaging. a** The relay is a fence. **b** The relay is a horizontal shutter. **c** The relay is an array of window edges. **d** The relay is a set of several sticks sparsely and randomly distributed.

introduce the virtual confocal signal **d** considered at the regular focal points $(x_i, y_j, 0)$, in which $i \in [I]$ and $j \in [J]$. The set of measurement pairs of the virtual confocal signal is denoted as $C_{virt} = \{(x_i, y_j, 0, x_i, y_j, 0) \mid i \in [I], j \in [J]\}$. The simulated signal generated with Eq. (2) at the set $C_{virt}$ is denoted by $A_\mathbf{d}\mathbf{u}$. The variable **d** is treated as an optimization variable and obtained together with the reconstruction by solving the optimization problem introduced in the next subsection. Let $C_{common} = C_{meas} \cap C_{virt}$, we denote by $R_\mathbf{b}(\mathbf{b}, \mathbf{d})$ the subset of **b** which is spatially located at the set $C_{common}$. We also write $R_\mathbf{d}(\mathbf{b}, \mathbf{d})$ the subset of the signal **d** which is considered at the set $C_{common}$. When $C_{common}$ is empty, both $R_\mathbf{b}(\mathbf{b}, \mathbf{d})$ and $R_\mathbf{d}(\mathbf{b}, \mathbf{d})$ are empty datasets.

**The Bayesian framework**

We treat the reconstructed target **u**, the measured signal $\tilde{\mathbf{b}}$, the approximated signal **b**, and the virtual confocal signal **d** as random vectors and formulate the imaging task as an optimization problem using Bayesian inference. The target and signals are obtained simultaneously by maximizing the joint posterior probability.

$$(\mathbf{u}^*, \mathbf{b}^*, \mathbf{d}^*) = \arg\max_{\mathbf{u}, \mathbf{b}, \mathbf{d}} \mathbb{P}(\mathbf{u}, \mathbf{b}, \mathbf{d} | \tilde{\mathbf{b}}) \quad (3)$$

Three assumptions are made to formulate this as a concrete optimization problem. Firstly, the conditional distribution of the measured signal $\tilde{\mathbf{b}}$ given the joint probability distribution of **u**, **b** and **d** is

$$\mathbb{P}(\tilde{\mathbf{b}} | \mathbf{u}, \mathbf{b}, \mathbf{d}) = \mathbb{P}(\tilde{\mathbf{b}} | \mathbf{u}, \mathbf{b}) = \exp(-|\mathbf{b} - \tilde{\mathbf{b}}|^2 - \Upsilon(\mathbf{u}, \mathbf{b}, \tilde{\mathbf{b}})) \quad (4)$$

in which $\Upsilon$ is related to the joint prior distribution of **u**, **b** and $\tilde{\mathbf{b}}$. With this assumption, **d** does not provide additional information to predict $\tilde{\mathbf{b}}$ when **b** is known. Secondly, the joint prior distribution of **u** and **b** is

$$\mathbb{P}(\mathbf{u}, \mathbf{b}) = \exp(-|A_\mathbf{b}\mathbf{u} - \mathbf{b}|^2 - \Gamma(\mathbf{u}, \mathbf{b})) \quad (5)$$

in which $\Gamma$ describes the prior distribution of **u** and **b**. The estimated signal **b** is less noisy than the measured data and is closer to the ideal signal of certain real-world targets, which helps to enhance the reconstruction quality. Thirdly, the conditional distribution of **d** given **u** and **b** is

$$\mathbb{P}(\mathbf{d} | \mathbf{u}, \mathbf{b}) = \exp(-|R_\mathbf{b}(\mathbf{b}, \mathbf{d}) - R_\mathbf{d}(\mathbf{b}, \mathbf{d})|^2 - |A_\mathbf{d}\mathbf{u} - \mathbf{d}|^2 - \Xi(\mathbf{u}, \mathbf{d})) \quad (6)$$

in which $R_\mathbf{b}(\mathbf{b}, \mathbf{d})$ and $R_\mathbf{d}(\mathbf{b}, \mathbf{d})$ are the subsets of the signals **b** and **d** that share the same measurement pairs. $\Xi(\mathbf{u}, \mathbf{d})$ is related to the joint prior distribution of the target **u** and the virtual confocal signal **d**.

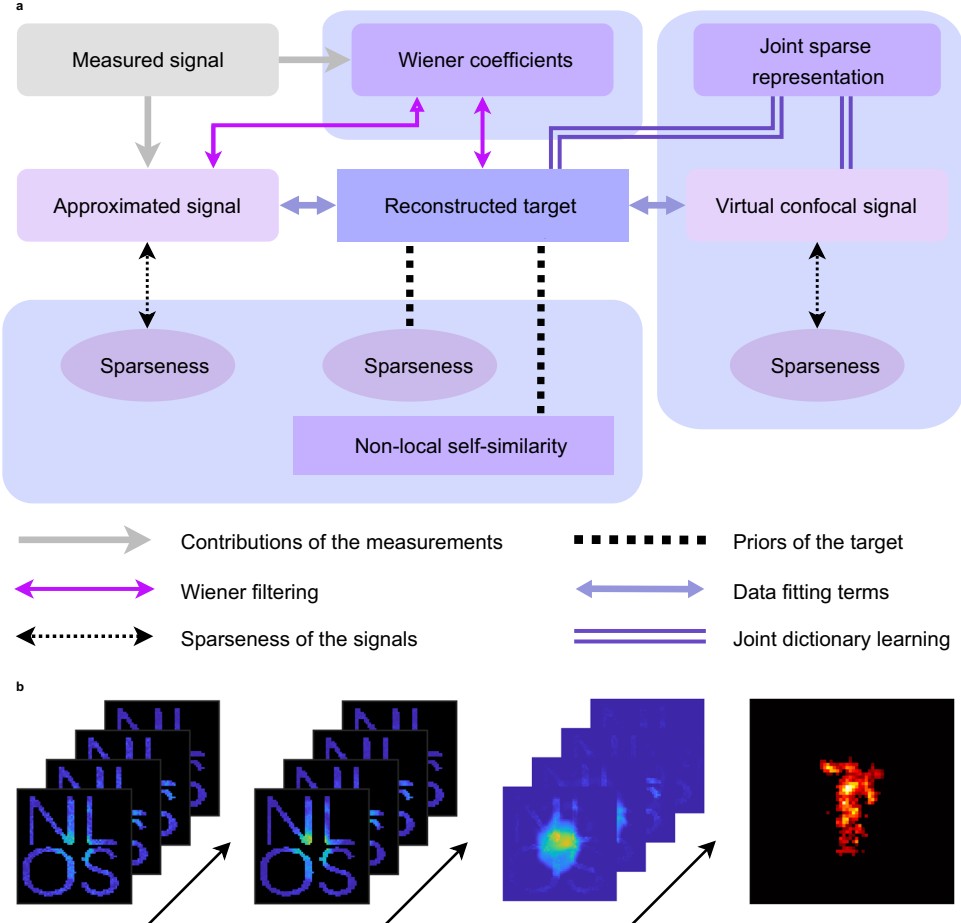

**Fig. 2 | The proposed CC-SOCR method. a** The CC-SOCR framework. For high quality reconstructions, the measured signal, estimated signal and the virtual confocal signal are treated as random variables and solved simultaneously using the Bayesian inference method. **b** The measured signal, the estimated signal, the virtual confocal signal and the reconstructed target are shown from left to right. The confocal measured data for the instance of the statue is provided in the Stanford dataset[8]. The relay region consists of four letters N, L, O, and S.

With these assumptions, we derive a concrete optimization problem using the Bayesian formula.

$$
\begin{aligned}
(\mathbf{u}^*,\mathbf{b}^*,\mathbf{d}^*) &= \arg\max_{\mathbf{u},\mathbf{b},\mathbf{d}} \mathbb{P}(\mathbf{u},\mathbf{b},\mathbf{d}|\tilde{\mathbf{b}}) \\
&= \arg\max_{\mathbf{u},\mathbf{b},\mathbf{d}} \mathbb{P}(\tilde{\mathbf{b}}|\mathbf{u},\mathbf{b},\mathbf{d})\mathbb{P}(\mathbf{u},\mathbf{b},\mathbf{d}) \\
&= \arg\max_{\mathbf{u},\mathbf{b},\mathbf{d}} \mathbb{P}(\tilde{\mathbf{b}}|\mathbf{u},\mathbf{b})\mathbb{P}(\mathbf{u},\mathbf{b},\mathbf{d}) \\
&= \arg\max_{\mathbf{u},\mathbf{b},\mathbf{d}} \mathbb{P}(\tilde{\mathbf{b}}|\mathbf{u},\mathbf{b})\mathbb{P}(\mathbf{d}|\mathbf{u},\mathbf{b})\mathbb{P}(\mathbf{u},\mathbf{b}) \\
&= \arg\min_{\mathbf{u},\mathbf{b},\mathbf{d}} |\mathbf{b}-\tilde{\mathbf{b}}|^2 + |R_{\mathbf{b}}(\mathbf{b},\mathbf{d})-R_{\mathbf{d}}(\mathbf{b},\mathbf{d})|^2 + |A_{\mathbf{d}}\mathbf{u}-\mathbf{d}|^2 \\
&\quad + |A_{\mathbf{b}}\mathbf{u}-\mathbf{b}|^2 + \Upsilon(\mathbf{u},\mathbf{b},\tilde{\mathbf{b}}) + \Xi(\mathbf{u},\mathbf{d}) + \Gamma(\mathbf{u},\mathbf{b})
\end{aligned}
\tag{7}
$$

in which the third equality follows from Eq. (4) and the last equality holds with Eqs. (4), (5) and (6). By designing appropriate regularization terms Y, $\Xi$ and $\Gamma$, we obtain high-quality reconstructions of the targets even in scenarios with highly incomplete measurements. The proposed framework and collaborative regularizations designed are illustrated in Fig. 2a. Concrete expressions of the regularizations are provided in the Methods section. We term the proposed method the confocal complemented signal-object collaborative regularization (CC-SOCR) due to the virtual confocal signal **d** introduced and the regularizations imposed on the signals and the target.

In the following, we compare the reconstruction results of the proposed method with the Laplacian of Gaussian filtered back-projection[33] (LOG-BP), F-K, LCT, PF and SOCR methods. For the LCT method, we adopt the D-LCT[31] extension that reconstructs both the albedo and surface normal. For the PF method, we adopt the implementation with the back-projection (PF-BP) algorithm[9] and the Rayleigh Sommerfeld Diffraction (PF-RSD) algorithm[32]. Performance comparisons of all these methods are shown in Table 1. To bring existing methods into comparisons in scenarios with incomplete measurements, we interpolate the signal with the nearest neighbor method[8,35], which generates better results than zero padding[32] (See Supplementary Fig. 24).

## Results on synthetic data

Instead of using an entire planar visible surface, we assume the relay to be a square box that simulates the scenario of the four edges of a window. The hidden object is a regular quadrangular pyramid, whose base length and height are 1 m and 0.2 m respectively. The central axis of the pyramid is perpendicular to the plane in which the relay square box lies, and the distance of the pyramid to this plane is 0.5 m. The albedo of the pyramid is assumed to be a constant. As shown in Fig. 3a, we simulate the signal measured at 36 points with Eq. (1). The points are exhaustively scanned, where only one point is illuminated each time, and signals are detected at all points. The dataset contains signals measured at 36 confocal and 1260 non-confocal pairs. The time resolution is set to 32 ps. Note that the LCT, D-LCT, F-K, PF-RSD and SOCR

**Table 1 | Comparisons of eight NLOS reconstruction algorithms**

| Method | Scenario | | Reconstructed target | | Reconstruction quality | |
|---|---|---|---|---|---|---|
| | Confocal measurements | Non-confocal measurements | Albedo | Surface normal | Dense measurements | Coarse measurements |
| LOG-BP[33] | General | General | ✓ | ✗ | Medium | Very Low |
| LCT[30] | Regular | ✗ | ✓ | ✗ | High | Low |
| D-LCT[31] | Regular | ✗ | ✓ | ✓ | High | Low |
| F-K[6] | Regular | ✗ | ✓ | ✗ | High | Low |
| PF-BP[9] | General | General | ✓ | ✗ | Medium | Low |
| PF-RSD[32] | Regular | Regular | ✓ | ✗ | High | Low |
| SOCR[34] | Regular | Regular | ✓ | ✓ | Very High | Medium |
| CC-SOCR | General | General | ✓ | ✓ | Very High | High |

By 'regular' we mean illumination and detection points uniformly distributed in a rectangular region. By 'general' we mean arbitrary illumination and detection points.

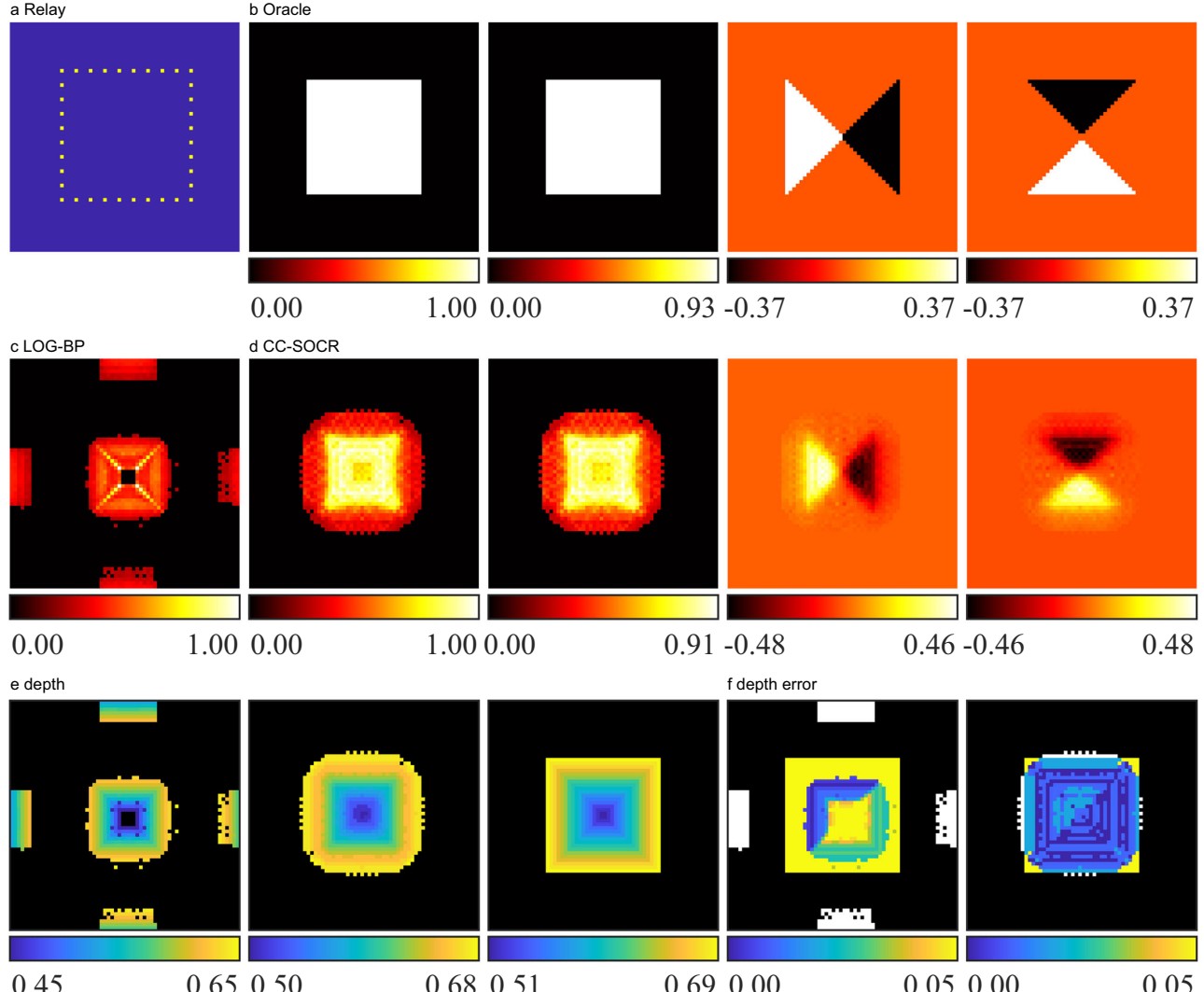

**Fig. 3 | Reconstruction results of the pyramid (non-confocal, synthetic signal).** **a** The illumination and detection points are shown in yellow. **b** Ground truth. The albedo as well as the depth, horizontal and vertical components of the directional albedo are shown from left to right. **c** The reconstructed albedo of the LOG-BP algorithm. **d** Reconstructed albedo and surface normal of the proposed CC-SOCR method. The albedo as well as the depth, horizontal and vertical components of the directional albedo are shown from left to right. **e** The depth of the LOG-BP, CC-SOCR reconstructions and the ground truth are shown from left to right. **f** The absolute depth error of the LOG-BP and CC-SOCR reconstructions are shown from left to right. The background is shown in black. Excessive voxels reconstructed are shown in white.

methods do not work directly in this scenario. We compare the reconstruction result of the proposed method with LOG-BP. The maximum intensity projections are shown in Fig. 3c and Fig. 3d. The reconstructed albedo is normalized to the range [0,1]. Albedo values that are less than 0.25 are thresholded to zero. The LOG-BP method fails to locate the target correctly and contains misleading artifacts near the boundary of the reconstruction domain. The proposed method locates the target correctly and does not contain noise in the background. The maximum depth error of the CC-SOCR reconstruction is 0.02 m, which is much smaller than the LOG-BP reconstruction

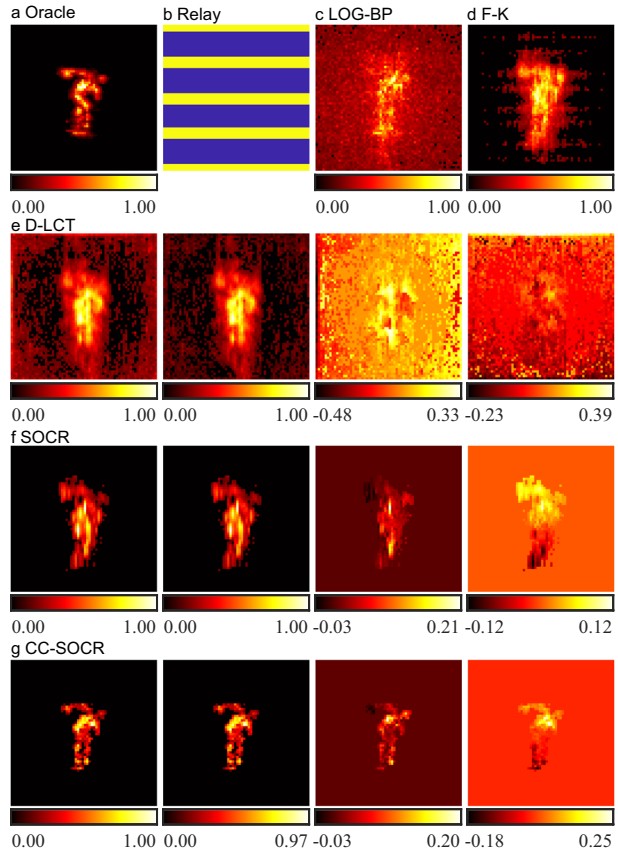

**Fig. 4 | Reconstructions of the statue with the relay surface in the shape of a horizontal shutter (confocal, measured signal). a** The oracle is generated with the SOCR method with $64 \times 64$ measurements[34]. **b** Confocal signals are measured in the yellow region. **c** Reconstructed albedo of the LOG-BP algorithm. **d** Reconstructed albedo of the F-K algorithm. **e** Reconstructed albedo and surface normal of the D-LCT method. The albedo as well as the depth, horizontal and vertical components of the directional albedo are shown from left to right. **f** Reconstructed albedo and surface normal of the SOCR method. The albedo as well as the depth, horizontal and vertical components of the directional albedo are shown from left to right. **g** Reconstructed albedo and surface normal of the CC-SOCR method. The albedo as well as the depth, horizontal and vertical components of the directional albedo are shown from left to right.

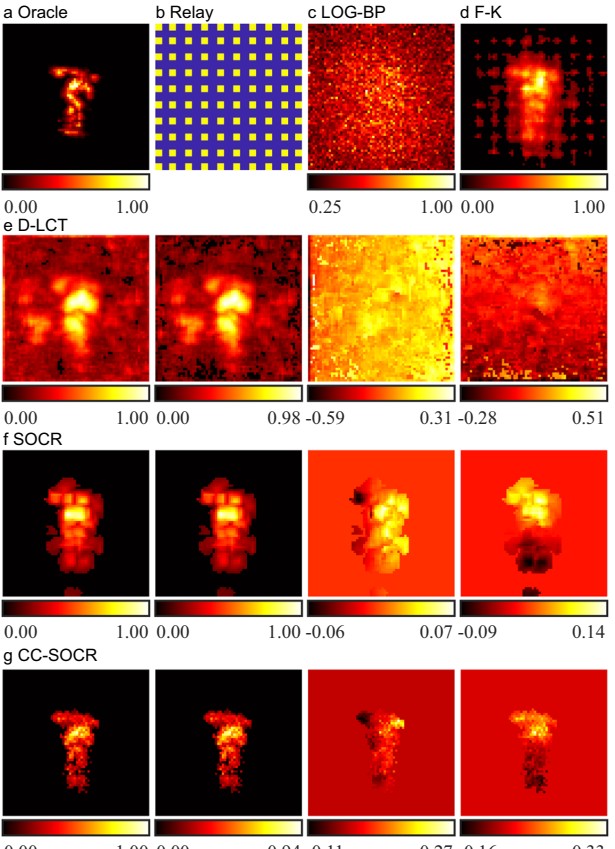

**Fig. 5 | Reconstructions of the statue with $10 \times 10$ confocal measurements (confocal, measured signal). a** The oracle is generated with the SOCR method with $64 \times 64$ measurements[34]. **b** Confocal signals are measured at the yellow points. **c** Reconstructed albedo of the LOG-BP algorithm. **d** Reconstructed albedo of the F-K algorithm. **e** Reconstructed albedo and surface normal of the D-LCT method. The albedo as well as the depth, horizontal and vertical components of the directional albedo are shown from left to right. **f** Reconstructed albedo and surface normal of the SOCR method. The albedo as well as the depth, horizontal and vertical components of the directional albedo are shown from left to right. **g** Reconstructed albedo and surface normal of the CC-SOCR method. The albedo as well as the depth, horizontal and vertical components of the directional albedo are shown from left to right.

(0.12 m). The absolute depth errors are shown in Fig. 3f. Classification error, defined as the percentage of excessive and missing voxels of the reconstruction, is used to assess the methods of locating the target. The classification error of the CC-SOCR reconstruction is 2.86%, which is nearly one order of magnitude smaller than that of the LOG-BP reconstruction (21.75%).

**Results on measured data**

For confocal experiments, we use the instance of the statue in the Stanford dataset[8] to test the performance of the proposed method. The target is 1 m away from the visible planar surface. In the original dataset, $512 \times 512$ focal points are raster-scanned in a square region of size $2 \times 2\,m^2$. The time resolution is 32 ps and the total exposure time is 60 min. An evenly distributed $64 \times 64$ dataset is sub-sampled from the original dataset, and it would take an exposure time of 56.25 s to measure this sub-sampled signal. The oracle shown in Fig. 4a and Fig. 5a are generated with the SOCR method using this sub-sampled signal. To simulate the case where the relay surface is a horizontal shutter, we only extract the signals measured at 21 rows from the downsampled data, as shown in the yellow region of Fig. 4b. From bottom to top, the five equispaced regions contain 3, 5, 5, 5 and 3 rows of measurements, respectively. The dataset contains signals measured

at 1344 focal points, which would take 18.46 s for data acquisition. Reconstruction results are shown in Fig. 4. The LOG-BP reconstruction is noisy. The reconstruction results of F-K, D-LCT and SOCR algorithms are blurry and contain artifacts. The proposed method reconstructs the target faithfully.

Figure 5 shows the reconstruction results of the statue with signals detected at $10 \times 10$ uniformly distributed focal points in a square region of size $2 \times 2\,m^2$, which would take 1.37 s for the measurements. The points scanned are shown in Fig. 5b. The LOG-BP reconstruction contains heavy background noise and the target cannot be clearly identified. The F-K and D-LCT reconstructions are blurry and also contain background noise. The SOCR reconstruction contains artifacts, indicating that the error of the signal introduced in the nearest neighbor interpolation process cannot be neglected. In contrast, the proposed method locates the target correctly and reconstructs more details than other methods. More reconstruction results with different numbers of uniformly distributed confocal measurements are compared in Supplementary Figs. 6–10.

Figure 6 shows the reconstruction results of the statue obtained with signals measured at different regions of the relay surface: a set of 200 randomly distributed focal points in an area of size $2 \times 2\,m^2$; a

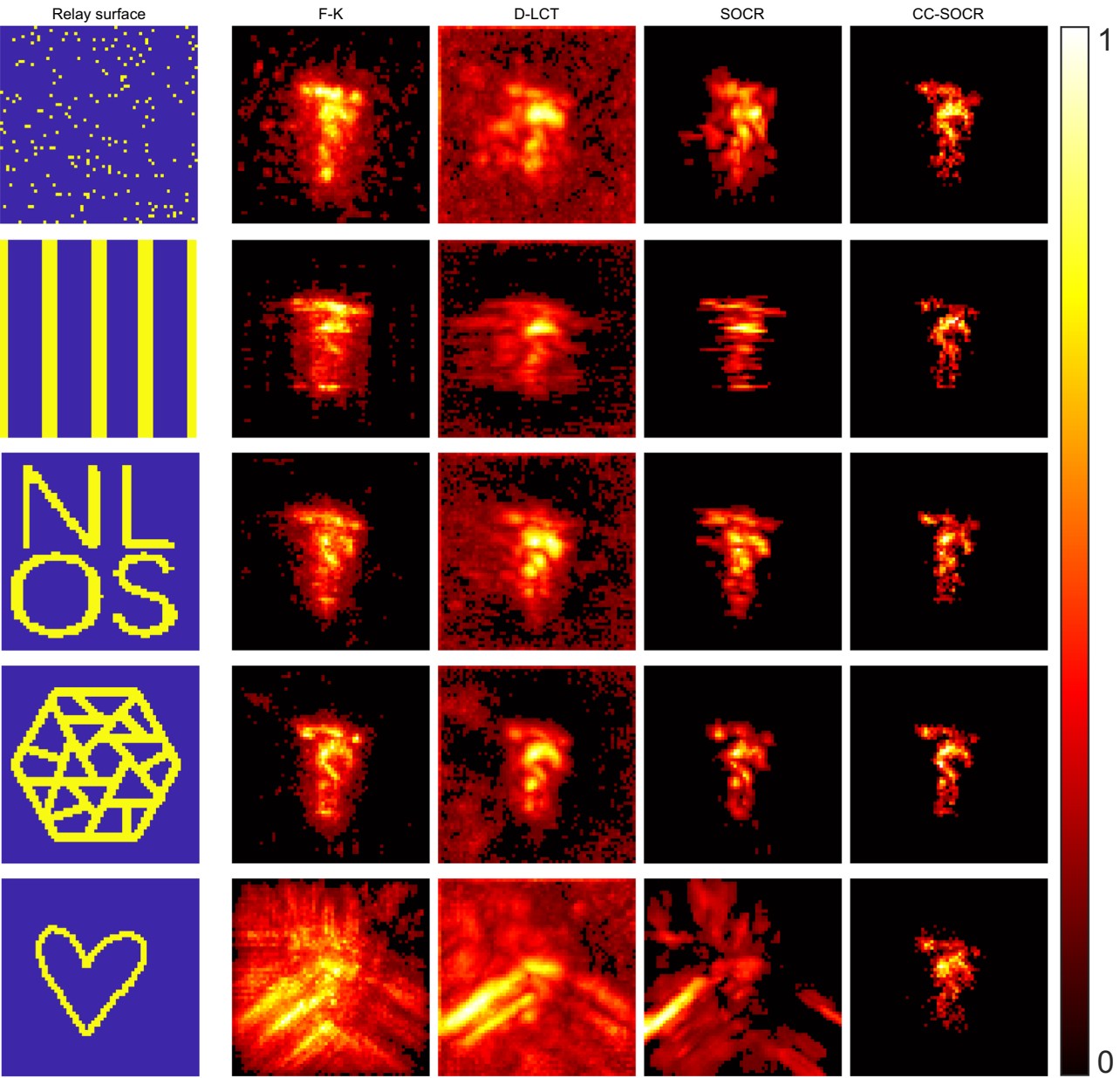

**Fig. 6 | Reconstructions of the statue under representative cases with different relays (confocal, measured signal).** The illumination regions are shown in yellow in the first column. The reconstructed albedo of F-K, D-LCT, SOCR and CC-SOCR methods are compared in the second to fifth columns.

region consisting of 5 equispaced vertical bars with 1344 focal points; a region that consists of four letters N, L, O and S with 825 focal points; a region made up of several sticks sparsely and randomly distributed with 1229 focal points; and a heart-shaped region with 258 focal points. These results indicate the capability of the proposed method in reconstructing the hidden target under various relay settings. For the case of the heart-shaped relay, the CC-SOCR method locates the target correctly, while all other methods fail. The measured signal, approximated signal and virtual confocal signal of the scenario with measurements at the four letters N, L, O and S are shown in Fig. 2b. The virtual confocal signal plays an important role for high-quality reconstruction. The three views and surface normal of the reconstructions as well as more comparisons under different relay settings are provided in Supplementary Figs. 11–17.

For non-confocal experiments, we use the measured data of the instance of the Figure 4 provided by the phasor field method[32]. The hidden object is 1 m away from the visible wall. The temporal

resolution is 16 ps. We pick out the signal measured at 64 × 64 illumination points in a square region of size 1.27 × 1.27 m². The detection point is 0.64 m to the left and 0.55 m to the bottom of the illumination region. Except for the signal selected, we also use four subsets of the signal to reconstruct the target: signals measured at five equispaced vertical bars that contain 3, 5, 5, 5, and 3 columns of focal points from left to right; signals measured at five equispaced horizontal bars that contain 3, 5, 5, 5, and 3 rows of focal points from bottom to top; signals measured at 14 × 14 uniformly distributed focal points in an area of 1.27 × 1.27 m²; signals measured at 200 randomly chosen focal points. To bring the PF-RSD and SOCR methods into comparison, the nearest neighbor interpolation technique is applied to extend the signal to 64 × 64 illuminations. As shown in Fig. 7, the LOG-BP and PF-BP reconstructions are noisy and contain artifacts. The PF-RSD reconstructions also contain artifacts. Both SOCR and CC-SOCR methods reconstruct the target successfully. However, the SOCR reconstructions contain artifacts (the third row) or lose some details (the fourth

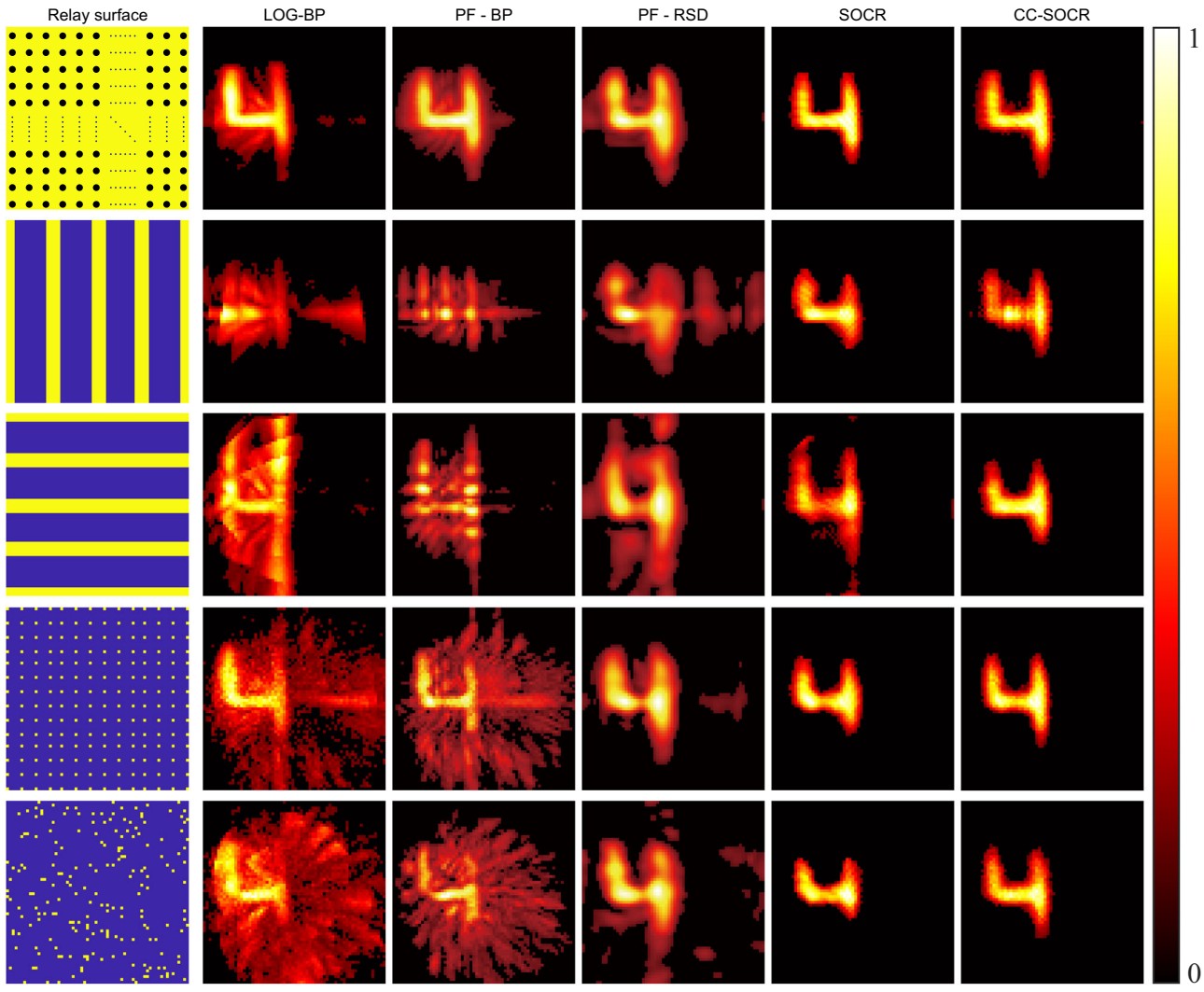

**Fig. 7 | Reconstruction results of the instance of the figure 4 (non-confocal, measured signal).** The illumination regions are shown in yellow in the first column. Reconstructed albedo of the LOG-BP, PF-BP, PF-RSD, SOCR and CC-SOCR methods are shown in the second to sixth columns.

and fifth rows). These results also indicate that the bias of the signal obtained from the nearest neighbor interpolation leads to non-negligible reconstruction error. The proposed CC-SOCR method provides faithful reconstructions in all cases. The three views and surface normal of the reconstructions are provided in Supplementary Figs. 18–22.

For scenarios with non-planar relay surfaces, we use the measured data in the Stanford dataset[8] to test the proposed method. The original dataset contains confocal signals measured at $128 \times 128$ focal points and is sub-sampled to $64 \times 64$. The NLOS scene contains two retroreflective letters, which leads to a bias with the physical model used. We extract subsets of the sub-sampled dataset to construct confocal and non-planar signals with irregular measurement patterns, as shown in opaque in the first column of Fig. 8. The proposed CC-SOCR method directly works under these settings and the results are shown in the last column. The LOG-BP method also works directly under these settings, but the reconstructions are of low quality and contain heavy background noise (See Supplementary Fig. 28). To bring the F-K, D-LCT and SOCR methods into comparisons, we shift the signal in the temporal dimension with the technique provided by the code of the F-K method. The shifted signals are then interpolated to $64 \times 64$ in spatial dimensions using the nearest neighbor method and serve as inputs of conventional imaging methods. As is shown in the last row of Fig. 8, the proposed method locates the targets

correctly with the oval-shaped non-planar illumination region, while all other methods fail.

## Discussion

We have proposed a framework for the general setting of NLOS imaging. In this section, we discuss its relationship with the original SOCR method, the complexity of the algorithm and possible directions for further improvements.

The SOCR method reconstructs the albedo and surface normal of the hidden targets under both confocal and non-confocal settings. However, the experimental setup is still quite limited. As demonstrated in the original paper[34], it only deals with signals measured at regular grid points. This is due to the spatial correlation of the signals in the regularization term.

The proposed CC-SOCR method generalizes the SOCR method to the most general setup, where no limitations of the measurement pairs are required. The CC-SOCR differs from SOCR in three aspects. Firstly, the introduced virtual confocal signal overcomes the rank deficiency of the measurement matrix, making it capable of reconstructing the targets under more general settings. Secondly, CC-SCOR does not include spatial correlations of the measured signal in the regularization term. As discussed in the Methods section, in CC-SOCR, the Wiener filter is applied only to the temporal dimension of the measured signal. Thirdly, the priors imposed on the target are related not

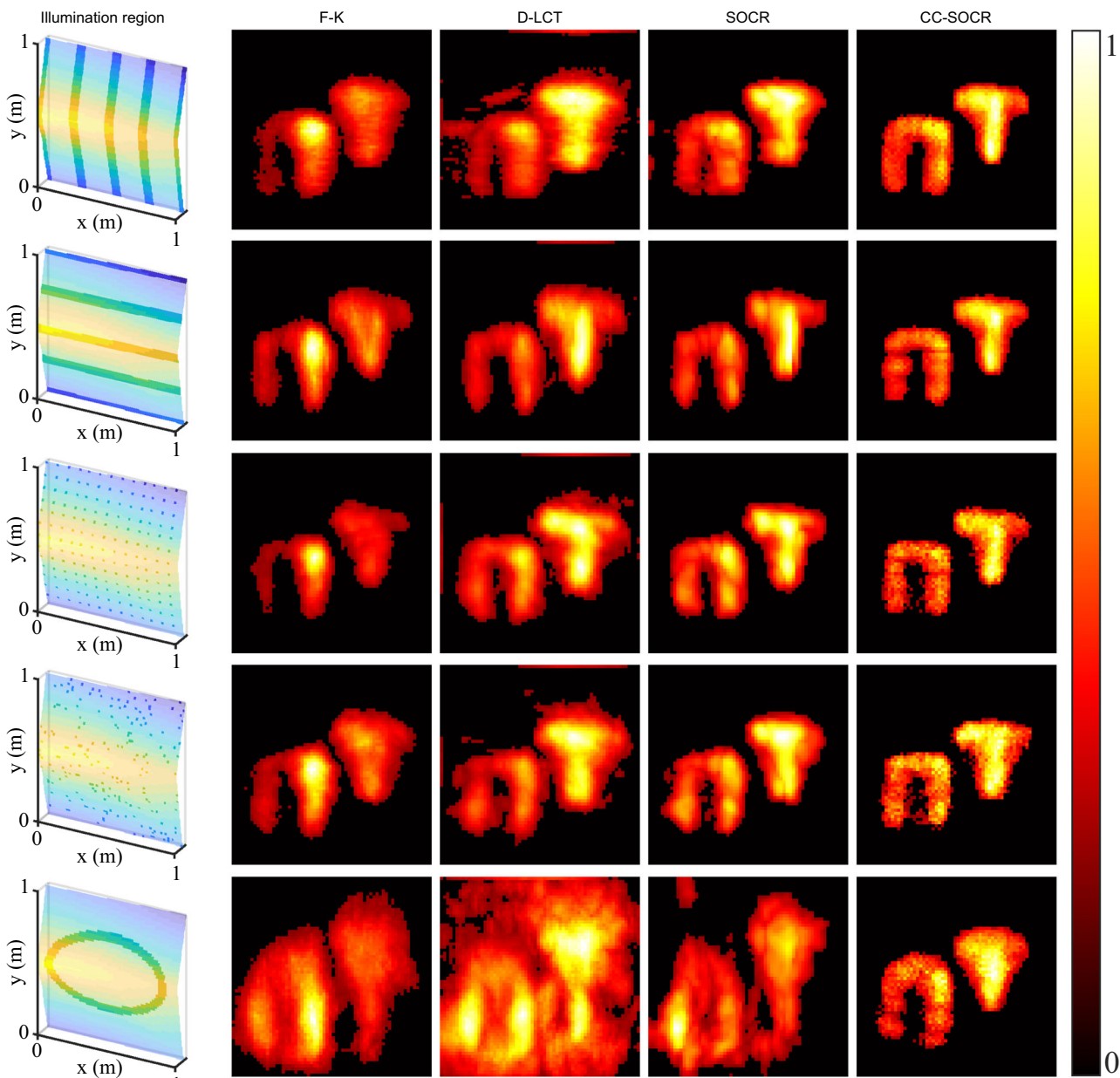

**Fig. 8 | Reconstructions of the letters N and T with irregular and non-planar relay settings (confocal, measured signal).** The illumination regions are shown in opaque in the first column. The F-K, D-LCT, SOCR and CC-SOCR reconstructions are shown in the second to the fifth columns, respectively.

only to the measured data but also to the introduced virtual confocal signal. Concrete expressions of these regularization terms are provided in the Methods section.

The proposed optimization problem can be solved efficiently using the alternative iteration method. In Supplementary Note 2, we decompose the problem into several sub-problems and discuss in detail the solutions to each sub-problem. We also provide a guide for choosing parameters in Supplementary Note 3. Convergences of all sub-problems are guaranteed, as discussed in the work of the SOCR method[34] and Supplementary Note 2. However, global convergence is not guaranteed because the sub-problem of updating the reconstructed target is solved approximately. Nonetheless, extensive results in Supplementary Note 1 have demonstrated the capability of the proposed method in providing high-quality reconstructions in various scenarios.

When the reconstruction domain is discretized with $N \times N \times N$ voxels and the signal is detected at $M$ measurement pairs, the memory complexity of the CC-SOCR algorithm is $O(\max\{N^3, MN\})$. The time complexity per iteration is $O(\max\{N^5, MN^3\})$, which is the same as the overall computation complexity. In Supplementary Note 4, we provide a detailed discussion of the complexity and report the running time for the instance of the statue with 200 randomly distributed confocal measurements. For the special case of $N \times N$ confocal measurements, the time and memory complexity is $O(N^5)$ and $O(N^3)$, which is the same as the SOCR algorithm. To reduce the computational complexity, the virtual confocal signal at coarser grids can be used. The time complexity reduces to $O(N^4)$ in scenarios with $O(N)$ measurement pairs if the virtual confocal signals are considered at $\sqrt{N} \times \sqrt{N}$ focal points. In Supplementary Figure 23, we compare the reconstruction results of the statue with virtual confocal signals of sizes $64 \times 64$, $32 \times 32$, $16 \times 16$ and $8 \times 8$ in an area of $2 \times 2\,m^2$, respectively. The execution time is provided in Supplementary Tables 3–6. Besides, the CC-SOCR algorithm can be implemented using the embarrassingly

parallel paradigm. The imaging process can be accelerated with GPU implementations of the code on large-scale parallel computing platforms. In the future, we would like to implement the octree representation of the reconstruction domain to reduce the complexity of the proposed method.

In CC-SOCR, virtual confocal signals observed at planar rectangular grid points are used to complement the reconstruction process in the case of incomplete measurements. It is also possible to consider virtual non-confocal signals for stronger regularizations. Besides, virtual confocal signals at several planes may be introduced to make use of the spatial correlation. However, the time and memory complexity will also increase.

With sufficient measurements, both the SOCR method and the CC-SOCR method provide high-quality reconstructions (See Supplementary Figs. 1, 11, 18). However, when the number of measurement pairs is small, the reconstruction problem is ill-posed. Although the complete signal can be obtained with interpolation techniques, existing methods still fail due to the bias introduced in the signal (Supplementary Fig. 27). The introduced virtual confocal signal benefits from the regularization guided by the simulated signal of the target and leads to faithful reconstructions. In the absence of the virtual confocal signal, the reconstructions may be blurry (Supplementary Fig. 25) or contain artifacts in the background (Supplementary Fig. 26). Besides, the CC-SOCR algorithm provides a robust way to convert measured non-confocal NLOS signals to their confocal counterparts. The generated confocal signal of the instance of figure 4 is provided in the supplementary code.

## Methods
### The joint regularizations
In Eq. (7), we formulate the CC-SOCR framework as an optimization problem. Here we show how the regularization terms $\Gamma(\mathbf{u},\mathbf{b})$, $\Xi(\mathbf{u},\mathbf{d})$ and $\Upsilon(\mathbf{u},\mathbf{b},\tilde{\mathbf{b}})$ are designed. To better grasp the idea of these regularization terms, we suggest a basic understanding of the data driven tight frame algorithm[40], the block matching and 3D filtering (BM3D) algorithm[41] and the SOCR method[34].

$\Gamma(\mathbf{u},\mathbf{b})$ describes the prior distribution of the reconstructed target and the approximated signal of the measurement pairs. For the reconstructed target, we consider the sparsity and non-local self-similarity priors and use the zero norm to impose sparseness on the approximated signal $\mathbf{b}$. We set

$$\Gamma(\mathbf{u},\mathbf{b}) = s_u|\mathbf{L}|_1 + \lambda_u \sum_i [|B_i(\mathbf{L}) - D_s C_i D_n^T|^2 + \lambda_{pu}|C_i|_0] + s_b|\mathbf{b}|_0 \quad (8)$$

in which $s_u, \lambda_u, \lambda_{pu}$ and $s_b$ are fixed parameters. $\mathbf{L}$ is the albedo of $\mathbf{u}$, $B_i$ is the block matching operator, with $i$ the index of a reference block. The summation is made over all possible blocks. $D_s$ and $D_n$ are two orthogonal matrices that capture the local structure and non-local correlations of the 3D albedo blocks. $C_i$ is the matrix consisting of transform coefficients of the $i^{th}$ block. $|\cdot|_0$ denotes the number of nonzero values of a tensor.

For the term $\Upsilon(\mathbf{u},\mathbf{b},\tilde{\mathbf{b}})$, we set

$$\Upsilon(\mathbf{u},\mathbf{b},\tilde{\mathbf{b}}) = \sum_i |P_i(\tilde{\mathbf{b}}) - DS_i|^2 + \sum_{i,j}\left(\frac{\sigma_{\mathbf{b}}}{d_j^T P_i(A_{\mathbf{b}}\mathbf{u})}S_i(j)\right)^2 \\ + \lambda_{sb}\sum_i |P_i(\mathbf{b}) - DS_i|^2 \quad (9)$$

in which $\lambda_{sb}$ is a fixed parameter, $P_i$ is the patch extracting operator, with $i$ the index of a local patch. Noting that the signals may not be measured at regular grid points, the patch extracting operator $P_i$ only applies to the temporal direction of the signals. $\tilde{\mathbf{b}}$ is the measured signal. $D$ is the matrix of discrete cosine transform. The $j^{th}$ filter of $D$ is denoted by $d_j$. $A_{\mathbf{b}}$ is the measurement matrix. $S_i$ is the vector that

consists of Wiener coefficients of the $i^{th}$ patch, with its $j^{th}$ element denoted by $S_i(j)$. $\sigma_{\mathbf{b}}$ is the noise level. The summations are made over all possible patches and filters of the discrete cosine matrix.

For the regularization term $\Xi(\mathbf{u},\mathbf{d})$, the prior of the virtual confocal signal $\mathbf{d}$ is constructed under the guidance of the target $\mathbf{u}$ and the physical model $A_{\mathbf{d}}$. Noting that the confocal signal $\mathbf{d}$ is considered at rectangular grid points, both the spatial and temporal correlations can be used. Let $P_i$ be the 3D patch extracting operator (2D in space and 1D in time), we seek a data-driven orthogonal dictionary $\Psi$ that sparsely represents the local patches of both the approximated signal $\mathbf{d}$ and the simulated signal $A_{\mathbf{d}}\mathbf{u}$. For simplicity, we abuse the notation $P_i$ to represent either a 1D patch of the measured signal $\mathbf{b}$ in the temporal direction or a 3D patch of the virtual confocal signal. The meaning can be made clear from the variable to which it applies. Let $Q_i$ be the matrix of transform coefficients of the $i^{th}$ patch, the regularization term is given by

$$\Xi(\mathbf{u},\mathbf{d}) = \sum_i [|Q_i - \Psi^T P_i(\mathbf{d})|^2 + \lambda_{sd}|Q_i - \Psi^T P_i(A_{\mathbf{d}}\mathbf{u})|^2 + \lambda_{fd}|Q_i|_0] + s_d|\mathbf{d}|_0 \quad (10)$$

in which $\lambda_{sd}$ and $\lambda_{fd}$ are two fixed parameters that control the weight of the simulated signal and the sparsity of the representation, respectively. $s_d$ is the parameter that controls the sparsity of the virtual confocal signal $\mathbf{d}$.

### The CC-SOCR optimization problem
By substituting Eqs. (8), (9), (10) into Eq. (7) and introducing weights, we obtain the concrete optimization problem of the proposed CC-SOCR framework as follows.

$$\min_{\mathbf{u},\mathbf{b},\mathbf{d},D_s,D_n,\mathbf{C},\mathbf{S},\Psi,\mathbf{Q}} |A_{\mathbf{b}}\mathbf{u} - \mathbf{b}|^2 + s_u|\mathbf{L}|_1 + s_b|\mathbf{b}|_0 \\ + \lambda_u \sum_i [|B_i(\mathbf{L}) - D_s C_i D_n^T|^2 + \lambda_{pu}|C_i|_0] \\ + \lambda_b|\mathbf{b} - \tilde{\mathbf{b}}|^2 + \lambda_b\lambda_{pb}\sum_i |P_i(\tilde{\mathbf{b}}) - DS_i|^2 \\ + \lambda_b\lambda_{pb}\sum_{i,j}\left[\frac{\sigma_{\mathbf{b}}}{d_j^T P_i(A_{\mathbf{b}}\mathbf{u})}S_i(j)\right]^2 \\ + \lambda_b\lambda_{pb}\lambda_{sb}\sum_i |P_i(\mathbf{b}) - DS_i|^2 \\ + \lambda_d|A_{\mathbf{d}}\mathbf{u} - \mathbf{d}|^2 + s_d|\mathbf{d}|_0 \\ + \lambda_d\lambda_{pd}\sum_i |Q_i - \Psi^T P_i(\mathbf{d})|^2 \\ + \lambda_d\lambda_{pd}\lambda_{sd}\sum_i |Q_i - \Psi^T P_i(A_{\mathbf{d}}\mathbf{u})|^2 \\ + \lambda_d\lambda_{pd}\lambda_{fd}\sum_i |Q_i|_0 \\ + \lambda_{bd}|R_{\mathbf{b}}(\mathbf{b},\mathbf{d}) - R_{\mathbf{d}}(\mathbf{b},\mathbf{d})|^2 \\ \text{s.t. } \mathbf{L} = \text{albedo}(\mathbf{u}), \\ D_s^T D_s = I[p_x p_y p_z], D_n^T D_n = I[r], \\ \Psi^T \Psi = I[q_x q_y q_t] \quad (11)$$

in which $\mathbf{C}$, $\mathbf{S}$ and $\mathbf{Q}$ represent the collections of the transform-domain coefficients $\{C_i\}$, $\{S_i\}$ and $\{Q_i\}$ respectively. $I[n]$ represents the identity matrix of order $n$. $p_x$, $p_y$ and $p_z$ are the patch sizes of the albedo in the horizontal, vertical and depth directions. $r$ is the number of neighboring blocks of each reference albedo block. $q_x$, $q_y$ and $q_t$ are the patch sizes of the virtual confocal signal $\mathbf{d}$ in the horizontal, vertical and temporal directions. $\sigma_{\mathbf{b}}$ is a parameter related to the noise level of the measured signal. The fixed parameters $s_u$, $s_b$, $s_d$, $\lambda_u$, $\lambda_b$, $\lambda_d$, $\lambda_{pu}$, $\lambda_{pb}$, $\lambda_{pd}$, $\lambda_{sb}$, $\lambda_{sd}$, $\lambda_{fd}$, $\lambda_{bd}$ balance the data-fitting terms and the regularization terms. The solution to the optimization problem is provided in Supplementary Note 2, and the supplementary software has been attached to this article.

## Data availability

The Zaragoza dataset is available in *Zaragoza NLOS synthetic dataset* [http://graphics.unizar.es/nlos_dataset.html]. The Stanford dataset can be downloaded at the project page [http://www.computationalimaging.org/publications/nlos-fk/]. The dataset provided by the phasor field method is available at the project page [https://biostat.wisc.edu/~compoptics/phasornlos20/fastnlos.html]. Synthetic data of the instance of the pyramid are attached to the code.

## Code availability

The code of the proposed method can be downloaded in the supplementary materials.

The accession codes of other methods are listed below.

"LOG-BP [https://springernature.figshare.com/articles/dataset/Datasets_and_reconstruction_code_for_a_virtual_wave_non-line-of-sight_imaging_approach/8084987]"

"F-K and LCT [http://www.computationalimaging.org/publications/nlos-fk/]"

"D-LCT [https://github.com/computational-imaging/nlos-dlct]"

"PF-RSD [https://biostat.wisc.edu/~compoptics/phasornlos20/fastnlos.html]"

"SOCR [https://www.nature.com/articles/s41377-021-00633-3]".

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

## Acknowledgements

This work was supported by the National Natural Science Foundation of China (61975087 F, 12071244 S, 11971258 Q).

## Author contributions
X.L. conceived the idea of the CC-SOCR method and implemented the code. X.L. and J.W. ran the experiments. L.X. created the art work of Fig. 1. Z.S. participated in the discussion of the Bayesian framework and the choice of parameters. X.F. initiated the idea of NLOS imaging with irregular illumination and detection pattern. L.Q. participated in the discussion of the solution to the proposed CC-SOCR optimization problem and suggested possible acceleration of the code. Z.S., X.F., and L.Q. supervised and directed this project. All authors discussed the results and contributed to the writing of the manuscript.

## Competing interests
The authors declare no competing interests.
