## [Peer Review File · Nature Communications]

Non-line-of-sight imaging with arbitrary illumination and detection patternREVIEWER COMMENTS

Reviewer #1 (Remarks to the Author):

The paper presents a method for active non-line-of-sight (NLOS) imaging from sparse measurements samples on a relay surface. While most NLOS imaging systems assume that the relay surface is densely sampled over some measurement area, this paper adds additional reconstruction priors to the recent Signal-Object Collaborative Regularization [34], improving robustness for sparse measurements. The paper thoroughly evaluates the method on simulated and captured datasets and shows some improvement compared to previous methods.

Still, there certainly tradeoffs to the method:

- First, the method seems to introduce significant practical complexity to the reconstruction approach. See, e.g., Eq. 11 which defines the objective function used in the optimization. The objective has 13 hyperparameters that need to be tuned and at least as many separate terms. This makes me wonder how much manual tuning is required to apply this method off the shelf to a new dataset.

- The method has a significant computational complexity compared to the conventional confocal methods ($O(N^5)$ vs $O(N^3)$). On one hand, the method can't exploit the regular measurement grids of the confocal measurements, so this makes sense; on the other hand given the sparse nature of the input measurements it's too bad that there's no computational improvement over previous work. Practically, the results therefore seem limited to low resolution.

There are also a number of related papers that address NLOS imaging for sparse measurements, limited relay surface, or dynamic/irregular relay surfaces. These are not currently discussed in the manuscript (though some are cited) and run counter to the claim in the paper that "this work demonstrates high quality NLOS reconstruction for the first time... with relay surfaces having discrete scattering regions, irregular shape, or very limited size...". I include some example references below.

- NLOS imaging on irregular grids/surfaces

Ref. [8] has example results and a method for an irregular relay surfaces.

La Manna, Marco, et al. "Non-line-of-sight-imaging using dynamic relay surfaces." Optics express 28.4 (2020): 5331-5339.

- NLOS imaging from limited size relay surface

Metzler, Christopher A., David B. Lindell, and Gordon Wetzstein. "Keyhole imaging: non-line-of-sight imaging and tracking of moving objects along a single optical path." IEEE Transactions on Computational Imaging 7 (2020): 1-12.

- NLOS imaging from sparse measurements

Isogawa, Mariko, et al. "Efficient non-line-of-sight imaging from transient sinograms." European Conference on Computer Vision. Springer, Cham, 2020.

I also have some technical questions below relating to the method and evaluation, detailed below.

- How exactly is \mathcal{D} generated? The text says that it is the orthogonal projection to a virtual planar surface perpendicular to the depth direction (L230), but what projection operation is used? On one hand the text says that \mathcal{D} is the confocal measurement pairs, but then it says that it is the projected voxels (presumably from the reconstruction domain). These definitions seem to be inconsistent.

- What is the R_b operator (L259)? Here, b and d seem to be of different dimensionality (since b are the measurements and d is the projected voxels. So I'm not sure how to understand this operation.

- Why is the Stanford dataset signal downsampled so much (to 64×64 from the original 512×512) and how is the downsampling performed? The downsampling should be performed using a subsampling operation rather than a true downsampling operation (i.e., blur and subsample) to simulate sampling the sparse locations on the wall.

- Phasor reconstructions are exactly the same in Supp. Figures 19-20. Is there a mistake there?

- I also wonder why the quality of Phasor fields is so low in the given examples. Is the nearest neighbor inpainting used with this method? Does the reconstruction quality improve with a different synthetic wavelength? I would have expected this method to be the most competitive given that the reconstructions don't rely on having a regular grid; instead, one can apply the backprojection/propagation operator only to the measured locations.

- Why are labels switched in Supp. Fig. 24 (top view vs side view on rows 1-3 and row 4)?

Reviewer #2 (Remarks to the Author):

The authors report impressive reconstruction results that improve upon the current state-of-the-art methods in the field of Non-Line-of-Sight (NLOS) imaging. Their CC-SOCR technique can reconstruct both the surface normal and the albedo of the hidden objects and can handle both confocal and non-confocal measurement schemes. This method builds on the SOCR method[1], which incorporates “prior knowledge” of signal sparsity and local surface orientation into a Bayesian framework to solve the inverse problem with high-quality reconstructions. However, the SOCR method requires a regular sampling grid on the relay surface as well as a large number of illumination/detection points on the relay wall.

The main novelty of the new CC-SOCR method comes from extending the Bayesian framework so that it is able to infer a virtual confocal signal from raw measurements. This enables the authors to deliver high-quality results for arbitrary illumination and detection patterns on the relay wall, at the expense of computational cost. Since NLOS data has been shown to benefit from using compressed sensing [2], the authors are able to maintain good quality reconstructions even when fewer points are sampled on the relay wall. This has the benefit of reducing restrictions on the geometry of the relay surface, number of scanning points, and total capture time. This has promising applications, especially in the fields of autonomous driving, surveillance, and disaster rescue.

I found this paper very interesting. This manuscript is well-written, and the reported results are convincing, on both experimental and synthetic datasets. Therefore, I recommend the publication of this work in Nature Communications, provided the authors are able to address my concerns below.

Major comments

1. The authors are claiming that their method can perform high-quality reconstructions “under the most general relay setting” (line 63). To the best of my knowledge, all the results shown are for scenes where the relay walls are 2D planar surfaces (although the spatial illumination and detection patterns on this planar surface have arbitrary shapes). Other methods have reported NLOS reconstruction for non-planar relay settings, [3], [4] and CC-SOCR shows some promise in generalizing to a non-planar surface since the virtual confocal signal is defined on a virtual plane. Therefore, I would like to see some results for non-planar relay walls that demonstrate this claim. If this method only works on planar relay walls, that should be made clear in the text, along with a brief discussion as to why this is the case.

Minor Revisions

Line 285/480: What is the justification for using nearest neighbors over a smarter 3D interpolation technique, such as cubic or spline interpolation, especially for confocal measurements? While I don't think a smarter interpolation technique will outperform this method, it would provide a fairer comparison.

Line 302: The methods listed here fail because they make use of Fast Fourier Transforms and require regular grid sampling on the relay wall. This also applies to PF when a fast RSD solver is used. [5] However, the phasor field framework may also be used with a slower backprojection algorithm [6], which is able to work directly on the scenario shown. My recommendation is to distinguish between the two cases, for example, by changing PF to "PF-RSD". Comparing the reconstruction with the slower PF method is not required.

Line 328: For clarity, please change "It takes 9 s to measure the downsampled signal." to "It would take an exposure time of 9s to measure this downsampled signal."

Line 389: Change "PF" to "PF-RSD". See my comment above for line 302.

Line 445: I believe the per iteration complexity is reported here. Even though the overall complexity will not change, it would be helpful to clarify this here.

Line 473: Please clarify what is meant by "sizes". The size of the virtual confocal signal could either refer to the grid spacing or the total field of view on the relay wall. I believe this refers to the grid spacing on the relay wall (while the total field of view of the wall is fixed) but it should be clarified. It is not clear from the supplement either.

General comments

1. It would be helpful to see the number of iterations used for SOCR and CC-SOCR methods in generating the results reported in Figs 3 – 7. The results reported are impressive, but I believe that computational bottlenecks will be an important consideration when attempting to decide which NLOS methods to deploy in the wild.
2. Due to the same reason, I would like to see a table (perhaps in the supplement) that lists the time and space complexity of all methods listed in table 1/table 2.
3. A secondary benefit of this method, in my opinion, is that it provides a more robust way to convert measured non-confocal NLOS signals to their confocal counterparts. Existing approaches, such as the normal moveout correction, are limited and can lead to artifacts in the reconstruction. [3] Therefore, these virtual confocal signals can serve as effective baselines to test future confocal methods on non-confocal datasets. I would like the authors to make available online the virtual confocal signals output by CC-SOCR for the non-confocal experimental dataset used in Figure 7.

References

- [1] X. Liu, J. Wang, Z. Li, Z. Shi, X. Fu, and L. Qiu, "Non-line-of-sight reconstruction with signal-object collaborative regularization," *Light Sci Appl*, vol. 10, no. 1, p. 198, Dec. 2021, doi: 10.1038/s41377-021-00633-3.
- [2] J.-T. Ye, X. Huang, Z.-P. Li, and F. Xu, "Compressed sensing for active non-line-of-sight imaging," *Opt. Express*, vol. 29, no. 2, p. 1749, Jan. 2021, doi: 10.1364/OE.413774.
- [3] D. B. Lindell, G. Wetzstein, and M. O'Toole, "Wave-Based Non-Line-of-Sight Imaging using Fast f-k Migration," vol. 38, no. 4, p. 13.
- [4] M. L. Manna, J.-H. Nam, S. A. Reza, A. Velten, and A. Velten, "Non-line-of-sight-imaging using dynamic relay surfaces," *Opt. Express, OE*, vol. 28, no. 4, pp. 5331–5339, Feb. 2020, doi: 10.1364/OE.383586.
- [5] X. Liu, S. Bauer, and A. Velten, "Phasor field diffraction based reconstruction for fast non-line-of-sight imaging systems," *Nature Communications*, vol. 11, no. 1, Art. no. 1, Apr. 2020, doi: 10.1038/s41467-020-15157-4.
- [6] X. Liu et al., "Virtual Wave Optics for Non-Line-of-Sight Imaging," p. 19.

Response to reviewers

We appreciate it very much for all the valuable comments from the reviewers, which greatly help to improve the paper. In this letter, we provide specific responses to these comments and list the corresponding modifications made to the manuscript. In the following, we term the proposed method the CC-SOCR method and Supplementary Algorithm 1 the CC-SOCR algorithm. The line numbers in the following refer to those in the document “revised manuscript without marking”.

Reviewer 1

1. Reviewer comment

The paper presents a method for active non-line-of-sight (NLOS) imaging from sparse measurements samples on a relay surface. While most NLOS imaging systems assume that the relay surface is densely sampled over some measurement area, this paper adds additional reconstruction priors to the recent Signal-Object Collaborative Regularization [34], improving robustness for sparse measurements. The paper thoroughly evaluates the method on simulated and captured datasets and shows some improvement compared to previous methods. Still, there certainly tradeoffs to the method:

Response

Thank you for your thorough review and constructive comments. We are glad to see that you found the method to be an improvement over previous methods and that you appreciate the thorough evaluation on simulated and captured datasets.

We agree that the assumption of densely sampled relay surfaces is common in NLOS imaging systems. We believe that our approach of adding additional reconstruction priors to the SOCR¹ provides improved robustness of the imaging system in the presence of fast sparse measurements. We also appreciate your suggestion to further discuss the tradeoffs of our method in the paper. In the following, we would like to discuss this of the proposed CC-SOCR method in detail.

2. Reviewer comment

First, the method seems to introduce significant practical complexity to the reconstruction approach. See, e.g., Eq. 11 which defines the objective function used in the optimization. The objective has 13 hyperparameters that need to be tuned and at least as many separate terms. This makes me wonder how much manual tuning is required to apply this method off the shelf to a new dataset.

Response

We thank the reviewer for pointing out this concern. The proposed CC-SOCR method involves many types of regularizations, and the corresponding optimization problem contains many parameters. However, as demonstrated in Supplementary Figures 17 and

26, these regularizations are necessary for high-quality reconstructions in scenarios with irregular measurement patterns.

We would like to discuss in detail the fixed parameters and the range of parameters that need manual tuning. The 14 parameters involved in the CC-SOCR optimization problem are s_u , s_b , λ_u , λ_{pu} , λ_b , λ_{pb} , σ_b , λ_{sb} , λ_d , s_d , λ_{pd} , λ_{sd} , λ_{fd} and λ_{bd} .

However, nine of them do not need manual tuning, as demonstrated in Supplementary Note 3 and the following:

- The parameters λ_b , λ_{pb} and λ_{sb} are fixed as 1, 16 and 0.25.
- The parameter λ_d used in step (1.5) is initialized as 2 (Eq. S6) and adaptively determined in step (2.3) with Eq. (S16), where λ_d^{imp} is fixed as 2.
- λ_{sd} and λ_{pd} are fixed as 1 and 4.
- The parameter λ_{bd} is fixed as 4.
- The parameter s_b is determined as $s_b = \lambda_b (s_b^{imp})^2$ and the parameter s_d is adaptively determined with Eq. S19, where s_b^{imp} and s_d^{imp} are set as 255×0.01 .

The parameters that need manual tuning are only two groups $\{\sigma_b, \lambda_{fd}, \lambda_{pu}\}$ and $\{s_u, \lambda_u\}$.

Group I: $\{\sigma_b, \lambda_{fd}, \lambda_{pu}\}$

The parameters σ_b and λ_{fd} are related to the noise level of the raw measurements, while the parameter λ_{pu} reflects the degree of noise reduction of the reconstructed target. There have been extensive studies on the estimation of noise levels²⁻⁴. In all experiments, these three parameters are chosen from the set $\{20, 40, 60\}$.

Group II: $\{s_u, \lambda_u\}$

The parameters s_u and λ_u control the weights of the sparsity and non-local self-

similarity regularizations of the target. The adaptive scheme of choosing s_u and an algorithmic parameter μ introduced for the split Bregman iteration have been studied¹. To determine λ_u , we choose λ_u^{imp} from the set $[0, 50]$ and compute λ_u with Eq. S15. The simpler geometric structures the target contains, the larger λ_u^{imp} should be used. The choice of s_u and λ_u can be determined at the end of step (1.2). One would inspect the sparse solution \mathbf{u}^0 to determine the sparsity of the scene and the complexity of the target.

These two groups of parameters used in all experiments are summarized in Table R1. The names of the parameters correspond exactly to those provided in the supplementary code.

Table R1 The parameters that need manual tuning in the CC-SOCR optimization problem

Parameters/Scenes	Bunny	Pyramid	Statue	Figure '4'	Letters 'N' and 'T'
sigma_b	60	40	60	40	60
lambda_fd_imp	60	40	60	40	20
lambda_pu_imp	40	60	40	60	40
s_u_imp	15	200	15	100	15
mu_imp	0.5	0.5	0.5	0.05	0.5
lambda_u_imp	5	15	5	10	50

Action taken

In the revised Supplementary Note 3, we have emphasized the parameters that need manual tuning and provided the range of these parameters. We have also added the following remarks.

Remark 1: For the instance of the pyramid, the parameters s_b^{imp} and s_d^{imp} are set as 255×0.001 because the generated simulated signal does not contain background noise.

Remark 2: In steps (1.3) and (2.4), the patch sizes and the searching window sizes for block matching and dictionary learning need manual tuning. Interested readers are referred to existing works^{1,5-7} for the choices of these parameters.

Remark 3: In steps (1.4) and (2.2), the patch size of the signal is fixed as $1 \times 1 \times 3$. In steps (1.6) and (2.6), the patch size of the signal is fixed as $3 \times 3 \times 3$.

Remark 4: For all experiments, the final results are obtained in no more than 2 iterations.

In steps (1.2) and (2.3), the conjugate gradient (CG) method is used to obtain the least-squares solutions. The maximum number of CG iterations is set to 20 and the iterations stop whenever the relative error of the normal equation reaches below 0.005. In these two sub-problems, the maximum numbers of the split Bregman iterations (Supplementary Algorithm 2) are fixed as 20 and 3 respectively. These iterations stop whenever the relative errors of the updated variables \mathbf{u} and \mathbf{v} reach below 0.005.

3. Reviewer comment

The method has a significant computational complexity compared to the conventional confocal methods ($O(N^5)$ vs $O(N^3)$). On one hand, the method can't exploit the regular measurement grids of the confocal measurements, so this makes sense; on the other hand given the sparse nature of the input measurements it's too bad that there's no computational improvement over previous work. Practically, the results therefore seem limited to low resolution.

Response

We agree with the reviewer that the computational complexity should be considered seriously in NLOS imaging. However, in certain scenarios with spatially irregular measurement patterns, the corresponding inverse problems are so ill-posed that all existing methods with low time complexity fail (See the last row of Figures 6, 8 and Supplementary Figures 4, 5, 17, 28).

In this work, we mainly focus on dealing with the ill-posedness of the problem and consider the time complexity as secondary. In our approach, the virtual confocal signal is introduced to deal with the rank deficiency of the measurement matrix, which takes $O(N^5)$ with the direct linear operator implementation of the physical model. Although the LCT, D-LCT, F-K and PF-RSD methods take $O(N^3 \log N)$, these methods may not work (Figure 3), fail to locate the hidden target (Figure 6), or provide low-quality reconstructions (Figures 4, 5, 6, 7, 8) under various settings of irregular measurement patterns. The proposed method stands out as the only one that correctly locates the targets in extreme cases (Supplementary Figures 4, 5, 17).

When the signals are detected at M pairs of illumination and detection points, and the reconstruction domain is discretized with $N \times N \times N$ voxels, the time and memory complexities of each step of the CC-SOCR algorithm with the $N \times N$ virtual confocal signal are summarized in Table R2.

Table R2 The time and memory complexities of the CC-SOCR algorithm

Step	Time complexity	Space complexity
(1.1)	$O(MN)$	$O(MN)$
(1.2)	$O(MN^3)$	$O(\max\{MN, N^3\})$
(1.3)	$O(N^3)$	$O(N^3)$
(1.4)	$O(MN^3)$	$O(MN)$
(1.5)	$O(N^5)$	$O(N^3)$
(1.6)	$O(N^5)$	$O(N^3)$
(2.1)	$O(MN^3)$	$O(\max\{MN, N^3\})$
(2.2)	$O(MN^3)$	$O(\max\{MN, N^3\})$
(2.3)	$O(\max\{MN^3, N^5\})$	$O(\max\{MN, N^3\})$
(2.4)	$O(N^3)$	$O(N^3)$
(2.5)	$O(N^5)$	$O(N^3)$
(2.6)	$O(N^5)$	$O(N^3)$
Total	$O(\max\{MN^3, N^5\})$	$O(\max\{MN, N^3\})$

In steps (1.5), (1.6), (2.5) and (2.6), generating the virtual confocal signal takes $O(N^5)$. In step (2.3), inverting the virtual confocal signal in the least-squares sense also takes $O(N^5)$. These are the computation bottlenecks of the CC-SOCR algorithm, which are resulted from the dense virtual confocal signal introduced.

We discuss two ways to reduce the computational complexity of the CC-SOCR method. Firstly, for sparse measurements with $O(N)$ pairs of illumination and detection points, the time complexity reduces to $O(N^4)$ if the virtual confocal signals at $N^{1/2} \times N^{1/2}$ focal points are used, but at the expense of sacrificing the reconstruction quality (See Supplementary Tables 3-6 and Supplementary Figure 23).

Secondly, for sparse measurements with $O(\log N)$ pairs of illumination and detection points, the time complexity can be further reduced to $O(N^3 \log N)$ with the $N \times N$ virtual confocal signal. To achieve this, the sub-problem of updating the reconstructed target (Step (2.3)) can be split further using the split Bregman method⁸. In this way, the contribution of the virtual confocal signal \mathbf{d} can be computed independently with the estimated signal \mathbf{b} . In this way, the process of generating and inverting the virtual confocal signal may be carried out using an extension of the D-LCT solver⁹, which takes $O(N^3 \log N)$. All other steps take no more than $O(N^3 \log N)$ and the overall time complexity remains $O(N^3 \log N)$. This might be a direction of future research.

Besides, the CC-SOCR algorithm can be implemented under the embarrassingly parallel paradigm. It is possible to accelerate the reconstruction process with GPU implementations of the code on large-scale parallel computing platforms.

As for the resolution of the reconstructed target, it could be very hard to obtain high-resolution results in scenarios with incomplete spatial measurements due to severe rank deficiency of the measurement matrix. Traditional methods may result in biased and

noisy reconstructions or fail to reconstruct the target even in low resolutions. The CC-SOCR method reconstructs the targets with spatial resolutions of 64×64 in horizontal and vertical directions.

Action taken

In the revised manuscript, we have deleted the sub-section “Virtual confocal signals at coarse grids” and added comments for the acceleration of the CC-SOCR algorithm in the sub-section “Time and memory complexities” to make the structure clearer.

4. Reviewer comment

There are also a number of related papers that address NLOS imaging for sparse measurements, limited relay surface, or dynamic/irregular relay surfaces. These are not currently discussed in the manuscript (though some are cited) and run counter to the claim in the paper that "this work demonstrates high quality NLOS reconstruction for the first time... with relay surfaces having discrete scattering regions, irregular shape, or very limited size...". I include some example references below.

- NLOS imaging on irregular grids/surfaces

Ref. [8] has example results and a method for an irregular relay surfaces.

La Manna, Marco, et al. "Non-line-of-sight-imaging using dynamic relay surfaces." Optics express 28.4 (2020): 5331-5339.

- NLOS imaging from limited size relay surface

Metzler, Christopher A., David B. Lindell, and Gordon Wetzstein. "Keyhole imaging: non-line-of-sight imaging and tracking of moving objects along a single optical path." IEEE Transactions on Computational Imaging 7 (2020): 1-12.

- NLOS imaging from sparse measurements

Isogawa, Mariko, et al. "Efficient non-line-of-sight imaging from transient sinograms." European Conference on Computer Vision. Springer, Cham, 2020.

Response

We thank the reviewer for reminding us of these works and would like to introduce them in the introduction section of the revised manuscript.

We would like to discuss the scope of application of these methods and the proposed method. The F-K method¹⁰ works in scenarios with planar or non-planar relay surfaces. In either case, dense measurements over a large square region are needed. The work of dynamic relay surface¹¹ uses the phasor field method to reconstruct the albedo of the hidden scene, which also depends on dense measurements in the entire relay region. The work of transient sinograms¹² requires circular and confocal measurements in a plane.

In these works, the illuminations are dense or enjoy certain geometric structures in spatial dimensions, which are not required in the CC-SOCR algorithm. To our best knowledge, only the back-projection(BP) type methods and the proposed one work under arbitrary illumination and detection patterns. However, the BP-type methods may fail due to incomplete data and heavy measurement noise. The work of keyhole imaging¹³ belongs to a different type which uses a single shot to track and reconstruct a moving hidden target. This method may fail when the target is still due to the ill-posedness of the reconstruction problem.

We agree with the reviewer that the claim “*This work demonstrates high quality NLOS reconstruction for the first time...*” is too strong and would like to revise this sentence.

Action taken

In the revised manuscript, we have added introductions for methods that deal with non-planar relay surfaces as follows. “For scenarios with non-planar relay surfaces, the F-K and back-projection type methods can be used directly. Algorithms designed only for planar relay settings can be applied using the signal shifting techniques^{10,11}.” We have also introduced the work of C²NLOS, compressed sensing, and keyhole imaging. “Isogawa et al. showed that the target could be reconstructed with confocal and circular NLOS scans¹². Sparse measurements from square grids scanning on the relay surface could also be used by incorporating the compressed sensing technique¹⁴. Besides, a single shot can be used to track a moving hidden target¹³, although the reconstruction fails when the target is still due to ill-posedness of the reconstruction problem.”

The sentence “*To the best of our knowledge, this work demonstrates high quality NLOS reconstruction for the first time...*” has been modified to “Our method demonstrates high-quality NLOS reconstructions in various scenarios with the relay surfaces having discrete scattering regions, arbitrary irregular shape, or very limited size, enabling the hidden object reconstruction with far more types of realistic relay surfaces such as window shutter, window frame, and fence, which significantly broadens the scope of NLOS imaging applications.”

5. Reviewer comment

I also have some technical questions below relating to the method and evaluation, detailed below.

How exactly is D generated? The text says that it is the orthogonal projection to a virtual planar surface perpendicular to the depth direction (L230), but what projection operation is used? On one hand the text says that D is the confocal measurement pairs, but then it says that it is the projected voxels (presumably from the reconstruction domain). These definitions seem to be inconsistent.

Response

We thank the reviewer for pointing out the ambiguity. We would like to explain the

definitions of D and \mathbf{d} using mathematical notations. Suppose that the reconstruction domain Ω is discretized with voxels $V = \{(x_i, y_j, z_k) | i \in [I], j \in [J], k \in [K]\}$, in which $[n] = \{1, 2, \dots, n\}$. x_i , y_j and z_k are the corresponding coordinates in the horizontal, vertical and depth directions, respectively. The set D is defined as

$$D = \{(x_i, y_j, 0) | i \in [I], j \in [J]\}$$

To overcome rank deficiency with spatially incomplete measurements, we consider the virtual confocal signals detected at the set D and denote the corresponding physical model as A_d (Eq. 2 in the manuscript). The simulated signal is denoted by $\mathbf{d}_{sim} = A_d \mathbf{u}$. The ideal signal \mathbf{d}_{ideal} is not known and we introduce an estimation \mathbf{d} of the ideal signal for the reconstruction. The estimated signal \mathbf{d} is treated as a random vector and obtained by solving the CC-SOCR optimization problem (Eq. 11 in the manuscript). The only input of this problem is the noisy measurement $\tilde{\mathbf{b}}$. The variables \mathbf{u} , \mathbf{b} , \mathbf{d} , D_s , D_n , \mathbf{C} , \mathbf{S} , Ψ and \mathbf{Q} are all obtained by solving the optimization problem.

Action taken

In the revised manuscript, we have replaced the sub-section “four types of signals” with two new sections “The measured signal” and “The virtual confocal signal”. For clarity, the spatial location of the virtual confocal signal has been introduced with mathematical notations. We have also emphasized that the variable \mathbf{d} is obtained by solving the CC-SOCR optimization problem.

6. Reviewer comment

What is the R_b operator (L259)? Here, b and d seem to be of different dimensionality (since b are the measurements and d is the projected voxels. So I'm not sure how to understand this operation.

Response

We thank the reviewer for pointing out the ambiguity and would like to make these

definitions clear. Consider a collection of M measurements. Let $p_m = (x_m^p, y_m^p, z_m^p)$ and $q_m = (x_m^q, y_m^q, z_m^q)$ be the coordinates of the m^{th} illumination and detection points. We call (p_m, q_m) a measurement pair. The set of coordinates of all measurement pairs is denoted by

$$C_{meas} = \{(x_m^p, y_m^p, z_m^p, x_m^q, y_m^q, z_m^q) | m \in [M]\},$$

and the corresponding estimated signal considered at C_{meas} is denoted by \mathbf{b} .

For the virtual confocal signal \mathbf{d} , the set of coordinates in spatial dimensions is given by

$$C_{virt} = \{(x_i, y_j, 0, x_i, y_j, 0) | i \in [I], j \in [J]\},$$

Now let $C_{common} = C_{meas} \cap C_{virt}$, we denote by $R_b(\mathbf{b}, \mathbf{d})$ the subset of \mathbf{b} which is spatially located at the set C_{common} . We also write $R_d(\mathbf{b}, \mathbf{d})$ the subset of the virtual confocal signal \mathbf{d} which is considered at the set C_{common} . Let $|C_{common}|$ be the number of elements of the set C_{common} , the variables $R_b(\mathbf{b}, \mathbf{d})$ and $R_d(\mathbf{b}, \mathbf{d})$ are both of size $|C_{common}| \times T$, where T is the maximum number of time bins used. When C_{common} is an empty set, both $R_b(\mathbf{b}, \mathbf{d})$ and $R_d(\mathbf{b}, \mathbf{d})$ are empty datasets.

Action taken

In the revised manuscript, we have made the definitions of $R_b(\mathbf{b}, \mathbf{d})$ and $R_d(\mathbf{b}, \mathbf{d})$ clear in the sub-section ‘‘The virtual confocal signal’’.

7. Reviewer comment

Why is the Stanford dataset signal downsampled so much (to 64x64 from the original 512x512) and how is the downsampling performed? The downsampling should be performed using a subsampling operation rather than a true downsampling operation (i.e., blur and subsample) to simulate sampling the sparse locations on the wall.

Response

We thank the reviewer for reminding us of the signal subsampling process. In the raster scanning mode, the exposure time is proportional to the number of measurement points, so fewer focal points are desirable. Besides, in the previous work¹, it was shown that

the statue in the Stanford dataset¹⁰ can be clearly reconstructed with 64×64 confocal measurements. In this work, we view this setting as an upper bound of the number of measurement pairs. We agree with the reviewer that the subsampling operator should be used. In the experiments, subsets of the original datasets are fetched and the subsampled dataset does not depend on signals detected at other points.

Action taken

We have emphasized that the subsampling operation is used in the revised manuscript.

8. Reviewer comment

Phasor reconstructions are exactly the same in Supp. Figures 19-20. Is there a mistake there?

Response

Yes. We thank the reviewer for pointing this out and apologize for this mistake. The phasor reconstruction of Supplementary Figure 19 is wrong.

Action taken

In the revised manuscript, we have replaced the phasor reconstruction of Supplementary Figure 19 with the correct one. We have also modified the term “Phasor” to “PF-RSD” for consistency with the main article (See also **Responses 9 and 14**). As is shown in Fig. R1, the format of the three views has also been adjusted for better visualization (See also **Response 10**).

9. Reviewer comment

I also wonder why the quality of Phasor fields is so low in the given examples. Is the nearest neighbor inpainting used with this method? Does the reconstruction quality improve with a different synthetic wavelength? I would have expected this method to be the most competitive given that the reconstructions don't rely on having a regular grid; instead, one can apply the backprojection/propagation operator only to the measured locations.

Response

We thank the reviewer for pointing out this concern. We would like to explain the reason for the low quality of phasor reconstructions and modifications made for fairer comparisons.

For the instance of figure ‘4’, the depth range of the reconstruction domain is fixed as 0.85 m ~ 1 m from the relay surface for fine visualizations of the three views of the reconstructions (Supplementary Figures 18 - 22). For the phasor field method, the size of the reconstructed albedo is $64 \times 64 \times 8$ in the horizontal, vertical and depth directions in the implementation with the Rayleigh Sommerfeld Diffraction (PF-RSD) algorithm. This method involves Fourier transform in the depth direction. However, the number of slices in the depth direction is so small that the high-frequency information is lost. This fact leads to blurry PF-RSD reconstructions. By extending the depth range of the reconstruction domain to 0.5 m ~ 1.5 m from the relay surface, the resolution of the reconstruction extends to $64 \times 64 \times 50$ and clearer PF-RSD reconstructions are obtained, as demonstrated below.

As suggested in the code of the phasor field method¹⁵, the synthetic wavelength is chosen as two times the sampling spacing multiplied by a sampling coefficient k . Another parameter that affects the reconstruction quality is the number of phasor wave cycles in the illumination pulse n . In the original manuscript, we fix $k = 2$ and $n = 3$. In the following two figures, we compare the PF-RSD reconstructions of the scene using the complete 64×64 illuminations with different settings of the depth ranges of the reconstruction domain, as well as different choices of the parameters k and n .

Fig. R3 PF-RSD reconstructions of the figure ‘4’ with the depth range of 0.5 m ~ 1.5 m from the relay surface. The reconstruction is of size $64 \times 64 \times 50$ in the horizontal, vertical and depth directions. The reconstructed albedo in the depth range of 0.85 m ~ 1 m is shown.

For better reconstruction quality, we would like to modify the depth range of the reconstruction domain to 0.5 m ~ 1.5 m and choose $k = 2$, $n = 4$ for the PF-RSD reconstructions. Only albedo values in the depth range of 0.85 m ~ 1 m will be illustrated for fair comparisons.

For scenarios with spatially incomplete measurements, we use the nearest neighbor inpainting technique to preprocess the signal to apply the PF-RSD algorithm. Besides, the PF method can also be implemented directly using the incomplete measurements with the back-projection (PF-BP) operator applied only to the measured locations. The

results are compared below, where the same reconstruction domain is illustrated.

Fig. R4 Phasor reconstructions of the figure ‘4’ with different relay settings and depth ranges. For the PF-RSD method, the signals are preprocessed with the nearest neighbor interpolation technique. The PF-BP method works directly under these settings, but the reconstructions contain artifacts.

As is shown in the figure above, the PF-BP reconstructions contain more artifacts than the PF-RSD method. For the best reconstruction quality, we would like to replace the phasor reconstructions with the results of the PF-RSD method with the depth range of the reconstruction domain chosen as 0.5 m ~ 1.5 m from the relay surface. The albedo values in the depth range of 0.85 m ~ 1 m will be illustrated for fair comparisons with other methods.

Action taken

In the revised manuscript, Figure 7 has been modified as follows.

Fig. R5 (This is also Fig. 7 in the revised manuscript.) Non-confocal reconstruction results of the instance of figure ‘4’ (non-confocal, measured signal). The illumination regions are shown in yellow in the first column. The reconstructed albedo of the LOG-BP, PF-RSD, SOCR and CC-SOCR are compared in the second to fifth columns.

Besides, in the revised Supplementary Figures 18 - 22, the corresponding PF-RSD reconstructions (sub-figures b.2) are modified as shown below.

Relay	Before revision			After revision		
	Front view	Top view	Side view	Front view	Top view	Side view
																														
Fig. R6 (These are also sub-figures b.2 in the revised Supplementary Figures 18 – 22.) The illumination regions are shown in yellow in the first column. The PF-RSD reconstructions after revision are less blurry than the original results.

10. Reviewer comment

Why are labels switched in Supp. Fig. 24 (top view vs side view on rows 1-3 and row 4)?

Response

We thank the reviewer for pointing this out. In the original Supplementary Figure 24, the corresponding labels were switched for white blanks that separate the sub-figures. We would like to adjust the formats and unify the orders of three views for all figures in the Supplementary Notes.

Action taken

In the revised Supplementary Figure 24, we have unified the order of three views as “front view”, “top view” and “side view”. As is shown in Fig. R7, we have also adjusted the formats for better visualization. All Supplementary Figures have been revised with the same format.

Reviewer 2

11. Reviewer comment

The authors report impressive reconstruction results that improve upon the current state-of-the-art methods in the field of Non-Line-of-Sight (NLOS) imaging. Their CC-SOCR technique can reconstruct both the surface normal and the albedo of the hidden objects and can handle both confocal and non-confocal measurement schemes. This method builds on the SOCR method[1], which incorporates “prior knowledge” of signal sparsity and local surface orientation into a Bayesian framework to solve the inverse problem with high-quality reconstructions. However, the SOCR method requires a regular sampling grid on the relay surface as well as a large number of illumination/detection points on the relay wall.

The main novelty of the new CC-SOCR method comes from extending the Bayesian framework so that it is able to infer a virtual confocal signal from raw measurements. This enables the authors to deliver high-quality results for arbitrary illumination and detection patterns on the relay wall, at the expense of computational cost. Since NLOS data has been shown to benefit from using compressed sensing [2], the authors are able to maintain good quality reconstructions even when fewer points are sampled on the relay wall. This has the benefit of reducing restrictions on the geometry of the relay surface, number of scanning points, and total capture time. This has promising applications, especially in the fields of autonomous driving, surveillance, and disaster rescue.

I found this paper very interesting. This manuscript is well-written, and the reported results are convincing, on both experimental and synthetic datasets. Therefore, I recommend the publication of this work in Nature Communications, provided the authors are able to address my concerns below.

Response

We thank the reviewer for the thorough reading and approval of our work. Besides, we also appreciate it very much for the valuable comment on non-planar experiments, which greatly helps enrich the reconstruction results of the proposed method. In the following, we show modifications made to address the concerns of the reviewer.

12. Reviewer comment

The authors are claiming that their method can perform high-quality reconstructions “under the most general relay setting” (line 63). To the best of my knowledge, all the results shown are for scenes where the relay walls are 2D planar surfaces (although the spatial illumination and detection patterns on this planar surface have arbitrary shapes). Other methods have reported NLOS reconstruction for non-planar relay settings, [3], [4] and CC-SOCR shows some promise in generalizing to a non-planar surface since the virtual confocal signal is defined on a virtual plane. Therefore, I would like to see some results for non-planar relay walls that demonstrate this claim. If this

method only works on planar relay walls, that should be made clear in the text, along with a brief discussion as to why this is the case.

[3] D. B. Lindell, G. Wetzstein, and M. O’Toole, “Wave-Based Non-Line-of-Sight Imaging using Fast f - k Migration,” vol. 38, no. 4, p. 13.

[4] M. L. Manna, J.-H. Nam, S. A. Reza, A. Velten, and A. Velten, “Non-line-of-sight-imaging using dynamic relay surfaces,” *Opt. Express, OE*, vol. 28, no. 4, pp. 5331–5339, Feb. 2020, doi: 10.1364/OE.383586.

Response

We thank the reviewer for reminding us of the non-planar experiments. We would like to use the measured confocal data in the Stanford dataset¹⁰ to test the proposed method with non-planar relay settings.

The original dataset contains confocal signals detected at 128×128 focal points and is sub-sampled to 64×64 in our experiments. The relay surface is shown in Fig. R8a. The NLOS scene contains two retroreflective letters, which leads to a bias with the physical model used in the CC-SOCR method. As is shown in Fig. R8c, the proposed method still correctly reconstructs the two letters with the sub-sampled dataset due to the strong regularizations introduced.

We further extract subsets of the sub-sampled dataset to construct confocal and non-planar signals with irregular measurement patterns, as shown in the first column of Fig. R8d. To bring the F-K, D-LCT and SOCR methods into comparisons, we shift the signal in the temporal dimension with the technique provided in the code of the F-K method¹⁰. The shifted signals are then interpolated to 64×64 in spatial dimensions using the nearest neighbor method and serve as inputs of conventional imaging methods. The proposed CC-SOCR method directly works under these settings and the results are shown in the last column of Fig. R8d. As is shown in the last row of Fig. R8d, the proposed method locates the targets correctly with the oval-shaped non-planar illumination region, while all other methods fail.

Fig. R8 (This is also Fig. 8 in the revised manuscript.) Reconstructions of the letters ‘N’ and ‘T’ with irregular and non-planar relay settings (confocal, measured signal). **a** The non-planar relay surface. **b** A photo of the targets. **c** The three views of the CC-SOCR reconstruction with 64×64 measurements. **d** Reconstruction results of the scene under different relay settings. The illumination regions are shown in opaque in the first column. The F-K, D-LCT, SOCR and CC-SOCR reconstructions are shown from the second to the fifth columns, respectively.

The LOG-BP method also works directly under these settings, but the reconstructions are of low quality and contain heavy background noise. The three views of the LOG-BP and CC-SOCR reconstructions are compared in Fig. R9.

Fig. R9 (This is also Supplementary Figure 28.) Reconstructions of the letters ‘N’ and ‘T’ with non-planar relay settings. a The illumination regions are shown in opaque. **b** The LOG-BP and CC-SOCR methods directly work under these relay settings. The three views of the reconstruction results with vertical, horizontal, sparse, random, and oval-shaped relay regions are shown from the first to the fifth rows, respectively.

Action taken

In the revised manuscript, we have added the non-planar NLOS reconstruction experiments mentioned above. We have also compared the three views of the LOG-BP and CC-SOCR reconstructions in Supplementary Figure 28.

13. Reviewer comment

Line 285/480: What is the justification for using nearest neighbors over a smarter 3D

interpolation technique, such as cubic or spline interpolation, especially for confocal measurements? While I don't think a smarter interpolation technique will outperform this method, it would provide a fairer comparison.

Response

We thank the reviewer for reminding us of smarter interpolation techniques. The linear, cubic and spline interpolations result in C^0 , C^1 , and C^2 continuity, respectively. In general, the transient images with complete spatial measurements are not continuous due to the delta functions contained in the forward model (Eq. 1 and 2 in the revised manuscript). The nearest neighbor interpolation technique provides discontinuous results and is easy to implement for scenarios with arbitrary irregular measurement patterns, so we use the nearest neighbor interpolation technique throughout the experiments.

For the instance of the statue with 10×10 confocal measurements, we show the reconstruction results of F-K, LCT, D-LCT and SOCR methods with the signal preprocessed with zero padding, nearest neighbor interpolation, linear interpolation, cubic interpolation and spline interpolation techniques in Fig. R10. It is shown that the target cannot be clearly reconstructed with these signal interpolation techniques. The zero padding method performs the worst, and other techniques also result in biased reconstructions of the target.

	F-K	LCT	D-LCT	SOCR
Zero padding				Nearest neighbor				Linear interpolation				Cubic interpolation				Spline interpolation				
Fig. R10 (This is also Supplementary Figure 27.) Reconstructions of the statue with 10×10 focal points and different signal interpolation techniques. The front view of the F-K, LCT, D-LCT and SOCR reconstructions are shown. The target cannot be clearly reconstructed with traditional signal interpolation techniques and conventional NLOS imaging algorithms.

The LOG-BP and the proposed CC-SOCR methods do not depend on the interpolation technique and the results are listed below for comparisons.

LOG-BP		CC-SOCR	--------	---	---------	---

Fig. R11 The LOG-BP and CC-SOCR reconstructions of the statue with 10×10 confocal measurements.

Action taken

We have added Fig. R10 as Supplementary Figure 27 to compare the zero padding,

nearest neighbor interpolation, linear interpolation, cubic interpolation and spline interpolation techniques for NLOS imaging with sparse confocal measurements.

14. Reviewer comment

Line 302: The methods listed here fail because they make use of Fast Fourier Transforms and require regular grid sampling on the relay wall. This also applies to PF when a fast RSD solver is used. [5] However, the phasor field framework may also be used with a slower backprojection algorithm [6], which is able to work directly on the scenario shown. My recommendation is to distinguish between the two cases, for example, by changing PF to “PF-RSD”. Comparing the reconstruction with the slower PF method is not required.

[5] X. Liu, S. Bauer, and A. Velten, “Phasor field diffraction based reconstruction for fast non-line-of-sight imaging systems,” Nature Communications, vol. 11, no. 1, Art. no. 1, Apr. 2020, doi: 10.1038/s41467-020-15157-4.

[6] X. Liu et al., “Virtual Wave Optics for Non-Line-of-Sight Imaging,” p. 19.

Response

We thank the reviewer for reminding us that PF-BP and PF-RSD methods should be distinguished. We also agree with the reviewer that the PF-RSD method be adopted in the manuscript because it generates better results (See also **Response 9**).

Action taken

We have emphasized that we adopt the implementation with Rayleigh Sommerfeld Diffraction (PF-RSD) solver for the phasor field method. We have also changed “PF” to “PF-RSD” in the revised manuscript. In the revised Supplementary Notes, the terms for the phasor field method have also been modified to PF-RSD for consistency with the main article.

15. Reviewer comment

Line 328: For clarity, please change “It takes 9 s to measure the downsampled signal.” to “It would take an exposure time of 9s to measure this downsampled signal.”

Response

We thank the reviewer for this helpful comment. We have checked the dataset again and found that the original 512×512 dataset takes an exposure time of 60 minutes. We apologize for this error and would like to revise the corresponding values and the syntax.

Action taken

In the revised manuscript, we have corrected the exposure time as “The time resolution is 32 ps and the total exposure time is 60 min.” We have also modified the corresponding sentence to “An evenly distributed 64×64 dataset is sub-sampled from the original dataset, and it would take an exposure time of 56.25 s to measure this sub-

sampled signal”.

16. Reviewer comment

Line 389: Change “PF” to “PF-RSD”. See my comment above for line 302.

Response

We thank the reviewer for pointing out the ambiguity.

Action taken

In the revised manuscript, we have changed “PF” to “PF-RSD”.

17. Reviewer comment

Line 445: I believe the per iteration complexity is reported here. Even though the overall complexity will not change, it would be helpful to clarify this here.

Response

We thank the reviewer for pointing this out. Indeed, the per-iteration complexity is the same as the overall complexity.

Action taken

In the revised manuscript, we have made it clear that the per-iteration complexity is reported and the overall complexity remains the same.

18. Reviewer comment

Line 473: Please clarify what is meant by “sizes”. The size of the virtual confocal signal could either refer to the grid spacing or the total field of view on the relay wall. I believe this refers to the grid spacing on the relay wall (while the total field of view of the wall is fixed) but it should be clarified. It is not clear from the supplement either.

Response

We thank the reviewer for pointing out the ambiguity. The sizes refer to the grid spacing on the relay wall and the total field of view is fixed as $2 \times 2 \text{ m}^2$. We would like to clarify this in the revised manuscript.

Action taken

We have modified the corresponding sentence to “In Supplementary Figure 23, we compare the reconstruction results of the statue with virtual confocal signals of sizes 64×64 , 32×32 , 16×16 and 8×8 in an area of $2 \times 2 \text{ m}^2$, respectively.”

19. Reviewer comment

It would be helpful to see the number of iterations used for SOCR and CC-SOCR methods in generating the results reported in Figs 3 – 7. The results reported are impressive, but I believe that computational bottlenecks will be an important consideration when attempting to decide which NLOS methods to deploy in the wild.

Response

We thank the reviewer for pointing this out. For the SOCR method, the number of iterations is the same as those provided in the code of the original article⁴. The number of iterations used for SOCR and CC-SOCR methods are summarized in the following table.

Table R3 The number of iterations used in SOCR and CC-SOCR methods

Figure	Instance	SOCR	CC-SOCR
Figure 1	Bunny	1	2
Figure 3	Pyramid	not applicable	2
Figure 4	Statue	1	2
Figure 5	Statue	1	2
Figure 6	Statue	1	2
Figure 7	Figure '4'	1	1
Figure 8	'N' and 'T'	1	2

Once the regularizations are applied, the SOCR and CC-SOCR algorithms converge quickly and further iterations do not bring much improvement.

Action taken

We have provided the supplementary code for the CC-SOCR algorithm, which includes the number of iterations used in all experiments.

20. Reviewer comment

Due to the same reason, I would like to see a table (perhaps in the supplement) that lists the time and space complexity of all methods listed in table 1/table 2.

Response

We agree with the reviewer that a table listing the complexity of all methods would be clear for comparisons. When the reconstruction domain is discretized with $N \times N \times N$ voxels and the signal is measured at M pairs of illumination and detection points ($M \leq N^2$), the time and space complexities are summarized in the following table. For the CC-SOCR method, we assume $N \times N$ virtual confocal signals.

Table R4 Comparisons of the time and space complexities of NLOS imaging algorithms with M measurement pairs and $N \times N \times N$ voxels

Method	Time complexity	Space complexity
LOG-BP ¹⁶	$O(MN^3)$	$O(N^3)$
LCT ¹⁶	$O(N^3 \log N)$	$O(N^3)$
D-LCT ¹⁷	$O(N^3 \log N)$	$O(N^3)$
F-K ¹⁰	$O(N^3 \log N)$	$O(N^3)$
PF-RSD ¹⁵	$O(N^3 \log N)$	$O(N^3)$
SOCR ¹	$O(N^5)$	$O(N^3)$
CC-SOCR	$O(\max\{MN^3, N^5\})$	$O(\max\{MN, N^3\})$

Besides, the time complexity of the CC-SOCR method reduces to $O(N^4)$ when $M = O(N)$ and the $N^{1/2} \times N^{1/2}$ virtual confocal signal is used (See also **Response 3**).

Action taken

In the revised Supplementary Notes, we have added a table (Supplementary Table 2) that lists the time and space complexities of all methods involved in the experiments.

21. Reviewer comment

A secondary benefit of this method, in my opinion, is that it provides a more robust way to convert measured non-confocal NLOS signals to their confocal counterparts. Existing approaches, such as the normal moveout correction, are limited and can lead to artifacts in the reconstruction. [3] Therefore, these virtual confocal signals can serve as effective baselines to test future confocal methods on non-confocal datasets. I would like the authors to make available online the virtual confocal signals output by CC-SOCR for the non-confocal experimental dataset used in Figure 7.

[3] D. B. Lindell, G. Wetzstein, and M. O'Toole, "Wave-Based Non-Line-of-Sight Imaging using Fast f - k Migration," vol. 38, no. 4, p. 13.

Response

We thank the reviewer for pointing out this additional benefit of our method. We would like to make the corresponding virtual confocal signals output by the CC-SOCR method publicly available online.

Action taken

We have added the benefit of converting the non-confocal NLOS signals to their confocal counterparts of the CC-SOCR algorithm in the discussion section. We have also provided the confocal signals of the instance of the figure '4' output by the CC-SOCR algorithm in the supplementary code.

Reference

1. Liu, X. *et al.* Non-line-of-sight reconstruction with signal–object collaborative regularization. *Light Sci. Appl.* **10**, 198 (2021).
2. Zoran, D. & Weiss, Y. From learning models of natural image patches to whole image restoration. in *2011 International Conference on Computer Vision* 479–486 (2011). doi:10.1109/ICCV.2011.6126278.
3. Chen, G., Zhu, F. & Heng, P. A. An Efficient Statistical Method for Image Noise Level Estimation. in *2015 IEEE International Conference on Computer Vision (ICCV)* 477–485 (2015). doi:10.1109/ICCV.2015.62.
4. Hou, Y. *et al.* NLH: A Blind Pixel-Level Non-Local Method for Real-World Image Denoising. *IEEE Trans. Image Process.* **29**, 5121–5135 (2020).
5. D Abov, K., Foi, A., Katkovnik, V. & Egiazarian, K. Image denoising with block-matching and 3D filtering. in *Proceedings of SPIE - The International Society for Optical Engineering* 354–365 (2006).
6. Lebrun, M. An Analysis and Implementation of the BM3D Image Denoising Method. *Image Process. Line* **2**, 175–213 (2012).
7. Cai, J.-F., Ji, H., Shen, Z. & Ye, G.-B. Data-driven tight frame construction and image denoising. *Appl. Comput. Harmon. Anal.* **37**, 89–105 (2014).
8. Goldstein, T. & Osher, S. The Split Bregman Method for L1-Regularized Problems. *SIAM J. Imaging Sci.* **2**, 323–343 (2009).
9. Young, S. I., Lindell, D. B., Girod, B., Taubman, D. & Wetzstein, G. Non-Line-of-Sight Surface Reconstruction Using the Directional Light-Cone Transform. in 1407–1416 (2020).

10. Lindell, D. B., Wetzstein, G. & O’Toole, M. Wave-based non-line-of-sight imaging using fast f - k migration. *ACM Trans. Graph.* **38**, 1–13 (2019).
11. La Manna, M., Nam, J.-H., Azer Reza, S. & Velten, A. Non-line-of-sight-imaging using dynamic relay surfaces. *Opt. Express* **28**, 5331 (2020).
12. Isogawa, M., Chan, D., Yuan, Y., Kitani, K. & O’Toole, M. Efficient Non-Line-of-Sight Imaging from Transient Sinograms. in *Computer Vision – ECCV 2020* (eds. Vedaldi, A., Bischof, H., Brox, T. & Frahm, J.-M.) vol. 12352 193–208 (Springer International Publishing, 2020).
13. Metzler, C. A., Lindell, D. B. & Wetzstein, G. Keyhole Imaging: Non-Line-of-Sight Imaging and Tracking of Moving Objects Along a Single Optical Path. *IEEE Trans. Comput. IMAGING* **7**, 12 (2021).
14. Xu, F., Ye, J., Huang, X. & Li, Z.-P. Compressed sensing for non-line-of-sight imaging. *Opt. Express* **29**, (2020).
15. Liu, X., Bauer, S. & Velten, A. Phasor field diffraction based reconstruction for fast non-line-of-sight imaging systems. *Nat. Commun.* **11**, 1645 (2020).
16. Matthew O’Toole, Lindell, D. B. & Wetzstein, G. Confocal non-line-of-sight imaging based on the light-cone transform. *Nature* **555**, 338–341 (2018).
17. Young, S. I., Lindell, D. B., Girod, B., Taubman, D. & Wetzstein, G. Non-Line-of-Sight Surface Reconstruction Using the Directional Light-Cone Transform. 10.

List of changes

1. We have added introductions of existing works for NLOS imaging with sparse measurements, as well as limited and dynamic relay surfaces. **(Lines 133 and 144)**
2. We have deleted the sub-section “Four types of signals” and added the sub-sections “The measured signal” and “The virtual confocal signal”. **(Lines 208 and 229)**
3. We have corrected the exposure time for the instance of the statue in the experiments. **(Lines 346, 348, 354 and 363)**
4. We have replaced the experiment of the instance of USAF resolution charts with the experiment using the measured non-planar and confocal signals. **(Line 424)**
5. We have added additional comments in the sub-section “Time and memory complexitiy” and deleted the sub-section “Virtual confocal signals at coarse grids”. **(Line 501)**
6. We have updated the experimental results for the PF-RSD reconstructions. **(Figure 7)**
7. We have corrected equation (10) by adding the L_0 term. **(Line 590)**

REVIEWERS' COMMENTS

Reviewer #1 (Remarks to the Author):

I'd like to thank the authors for their detailed responses to the reviewer comments.

I especially appreciated the clarifications about hyperparameters and computational complexity. The method does require a significant amount of computation for reconstructing relatively low resolution hidden scenes (I tried running the included code, which did work as expected). Still, the reconstructions from sparse measurements show improvement over what can be done using existing methods.

My main concern remains to be about the complexity of the method (e.g., the 14 hyperparameters, of which 5 still require manual tuning). The rebuttal shows that each listed scene uses different hyperparameters (Table R1), and I expect that selecting these hyperparameters for each scene is tedious given the long optimization times.

Also, I think the complexity of the algorithm yet makes the text difficult to parse at places. For example, consider Eq. 11: I'd expect most readers to have a difficult time parsing this objective function (the mathematical expression takes up half a page with just under 20 terms).

Unfortunately, I don't have any concrete suggestions about how to improve the complexity of the method or clarity of the text beyond somehow simplifying the method, if that is even possible. One might try to simplify Eq 11 and its corresponding version in the supplement by grouping terms under new symbols that are defined beforehand and correspond to penalties/regularization terms that are conceptually similar (but this adds even more symbols to an already substantial number).

On a positive note, the inclusion of pseudocode, code, and data in the supplement is very helpful and should aid reproducibility.

Beyond this, my other questions were answered in the rebuttal and I have no other technical criticisms of the paper.

Reviewer #2 (Remarks to the Author):

Authors have done a good job of addressing my concerns and demonstrated the utility of their algorithm on a non-planar surface, which greatly improves the quality of this manuscript. While their results are indeed impressive, the non-planar surface used in their example (and in the original f-k migration paper) is fairly similar to a planar one and uses retroreflective targets in the hidden scenes. It is not clear to me how this algorithm will perform on general surfaces with arbitrary curvature, diffuse hidden targets, and non-confocal configurations, but this can perhaps be the subject of future work. Due to this, I recommend changing the text “under the most general setting” in line 162 so it does not mislead people unfamiliar with the field.

I appreciate the thorough comparisons in Response 9, and I recommend adding PF-BP results from Fig. R4 as a column to Figure 7 since the results have already been generated. Also, Figure 7 in the final manuscript needs to be updated as the PF column is the same as before.

A few typos:

Line 63: change “setting” to settings

Line 178 – 180: Add commas to this sentence for clarity, so it reads as “Reconstruction results of the bunny with synthetic confocal signals, detected at the entire relay surface and these four scenarios, are provided in Supplementary Figures 1 – 5.”

Line 185: Change “these signals” to “these measured signals” for clarity

The authors have addressed all my concerns in a satisfactory manner. With these revisions, I recommend that the article be published in Nature Communications.

Response to reviewers

We thank the reviewers for the thorough review. In this letter, we provide specific responses to the comments and list the corresponding modifications made to the manuscript. In the following, we term the proposed method the CC-SOCR method and Supplementary Algorithm 1 the CC-SOCR algorithm. The line numbers in the following refer to those in the document “revised manuscript without marking”.

Reviewer 1

1. Reviewer comment

I'd like to thank the authors for their detailed responses to the reviewer comments. I especially appreciated the clarifications about hyperparameters and computational complexity. The method does require a significant amount of computation for reconstructing relatively low resolution hidden scenes (I tried running the included code, which did work as expected). Still, the reconstructions from sparse measurements show improvement over what can be done using existing methods.

Response

We thank the reviewer for checking our code and acknowledging the results provided by the proposed method. Indeed, the CC-SOCR method brings additional computational cost due to the virtual confocal signal introduced. In the current stage, we aim at providing high-quality reconstructions for NLOS detection scenarios with arbitrary illumination and detection patterns, which is a non-trivial task due to the ill-posedness nature of the inverse problem. We would like to reduce the complexity of the CC-SOCR method in the future.

2. Reviewer comment

My main concern remains to be about the complexity of the method (e.g., the 14 hyperparameters, of which 5 still require manual tuning). The rebuttal shows that each listed scene uses different hyperparameters (Table R1), and I expect that selecting these hyperparameters for each scene is tedious given the long optimization times.

Response

We thank the reviewer for pointing out this concern and would like to discuss more about the 5 parameters that need manual tuning.

The parameter s_u can be determined in step (1.2) of the CC-SOCR algorithm (**Supplementary Algorithm 1**), which is at the early stage of the whole algorithm. In our setting, the reconstruction domain is discretized with 3D voxels, but we are interested in reconstructing the surface of the NLOS target where photons are bounced back. Thus, a majority of the voxels of the directional albedo should be zero. It is possible to design self-adaptive schemes to determine the parameter s_u based on the

number of non-zero elements of the sparse reconstruction \mathbf{u}^0 , but at the expense of solving the sub-problem (1.2) several times.

The parameter λ_u controls the weight of the dictionary learning term of the albedo. In our experience, $\lambda_u^{imp} = 10$ usually yields good results.

The parameters σ_b , λ_{fd} and λ_{pu} correspond to noise levels and can be chosen from the set $\{20, 40, 60\}$. We suggest fixing these parameters as 40, which generates good results in most experiments.

We also remark that the parameters are not sensitive to the illumination and detection patterns for fixed NLOS scenes. In our experiments, the parameters are fixed for different instances of the measurement patterns for each NLOS scene.

To reduce the time complexity, one might introduce the octree representation of the reconstruction domain. To filter out the noise of the reconstruction, the algorithm of density based clustering of application with noise (DBSCAN) might be used. We would like to try these methods in the future.

3. Reviewer comment

Also, I think the complexity of the algorithm yet makes the text difficult to parse at places. For example, consider Eq. 11: I'd expect most readers to have a difficult time parsing this objective function (the mathematical expression takes up half a page with just under 20 terms).

Response

We agree with the reviewer that the proposed optimization problem can be difficult to parse. However, in certain scenarios (for example, the heart-shaped relay for the instance of the statue), only the CC-SOCR method reconstructs the shape of the target successfully. This demonstrates the necessity of introducing the virtual confocal signal and the regularization terms. For a better understanding of the regularization terms introduced, we would like to add some references to guide the understanding of the regularization terms.

Action taken

In the **Methods** section, we have added the sentence *“To better grasp the idea of these regularization terms, we suggest a basic understanding of the data driven tight frame algorithm¹, the block matching and 3D filtering (BM3D) algorithm² and the SOCR method³”* (Line 538)

4. Reviewer comment

Unfortunately, I don't have any concrete suggestions about how to improve the complexity of the method or clarity of the text beyond somehow simplifying the method, if that is even possible. One might try to simplify Eq 11 and its corresponding version in the supplement by grouping terms under new symbols that are defined beforehand and correspond to penalties/regularization terms that are conceptually similar (but this adds even more symbols to an already substantial number).

Response

We agree with the reviewer that simplification of the objective function would help to understand the method. For simplicity, we have tried to split the final objective function to contain as many isolated terms as possible from the original definition of the regularization terms. We also agree with the reviewer that replacing some of the regularization terms with conceptually similar and numerical friendly ones would help to reduce the complexity. Currently, solving L_1 regularized problems takes iterations and can be slow in practice. However, the L_1 regularization term does improve the reconstruction quality significantly. In the future, we would like to develop other regularization terms to reduce the time complexity of the CC-SOCR method while maintaining the reconstruction quality.

5. Reviewer comment

On a positive note, the inclusion of pseudocode, code, and data in the supplement is very helpful and should aid reproducibility. Beyond this, my other questions were answered in the rebuttal and I have no other technical criticisms of the paper.

Response

We thank the reviewer again for the comprehensive review of our manuscript and would like to improve the issues mentioned above in future works.

Reviewer 2

6. Reviewer comment

Authors have done a good job of addressing my concerns and demonstrated the utility of their algorithm on a non-planar surface, which greatly improves the quality of this manuscript. While their results are indeed impressive, the non-planar surface used in their example (and in the original $f-k$ migration paper) is fairly similar to a planar one and uses retroreflective targets in the hidden scenes. It is not clear to me how this algorithm will perform on general surfaces with arbitrary curvature, diffuse hidden targets, and non-confocal configurations, but this can perhaps be the subject of future work. Due to this, I recommend changing the text "under the most general setting" in line 162 so it does not mislead people unfamiliar with the field.

Response

We thank the reviewer for acknowledging the value of our work, and agree with the reviewer that the claim “under the most general setting” is too strong. The proposed CC-SOCR method works for arbitrary illumination and detection pattern and we would like to carry out more experiments with non-planar relay surfaces in the future.

Action taken

In the revised manuscript, we have changed the sentence “*The proposed method works very well under the most general setting, allowing regular and irregular measurement patterns in both confocal and non-confocal scenarios.*” to “*The proposed method allows regular and irregular measurement patterns in both confocal and non-confocal scenarios.*” (Line 161)

7. Reviewer comment

I appreciate the thorough comparisons in Response 9, and I recommend adding PF-BP results from Fig. R4 as a column to Figure 7 since the results have already been generated. Also, Figure 7 in the final manuscript needs to be updated as the PF column is the same as before.

Response

We thank the reviewer for pointing out that adding PF-BP results from Fig. R4 as a column to Figure 7 would enrich the comparison. We also thank the reviewer for reminding us that Figure 7 in the manuscript should be updated. In the revised manuscript, Figure 7 is updated as shown below.

Fig. RR1 (This is also Fig. 7 in the revised manuscript.) Reconstruction results of the instance of figure 4 (non-confocal, measured signal). The illumination regions are shown in yellow in the first column. Reconstructed albedo of the LOG-BP, PF-BP, PF-RSD, SOCR and CC-SOCR methods are shown in the second to sixth columns.

Action taken

In the revised manuscript, we have included the PF-BP reconstructions for comparisons in Figure 7. Corresponding statements have also been revised. (Lines 297 and 410, Figure 7 and Table 1)

8. Reviewer comment

A few typos:

Line 63: change “setting” to settings

Line 178 – 180: Add commas to this sentence for clarity, so it reads as “Reconstruction results of the bunny with synthetic confocal signals, detected at the entire relay surface and these four scenarios, are provided in Supplementary Figures 1 – 5.”

Line 185: Change “these signals” to “these measured signals” for clarity

Response

We thank the reviewer for pointing out the typos and would like to revise the corresponding sentences.

Action taken

In the revised manuscript, the corresponding sentences have been revised as follows.

“Our approach is capable of reconstructing both the albedo and surface normal of the hidden objects with fine details under general relay settings.” (Line 63)

“Reconstruction results of the bunny with synthetic confocal signals, detected at the entire relay surface and these four scenarios, are provided in Supplementary Figures 1 – 5.” (Line 179)

“The goal of NLOS imaging is to take a collection of measured transient data and find the target that comes closest to fitting these measured signals.” (Line 184)

9. Reviewer comment

The authors have addressed all my concerns in a satisfactory manner. With these revisions, I recommend that the article be published in Nature Communications.

Response

We would like to thank the reviewer again for the comprehensive review of our manuscript, which significantly helps to improve our work.

References

1. Cai, J.-F., Ji, H., Shen, Z. & Ye, G.-B. Data-driven tight frame construction and image denoising. *Appl. Comput. Harmon. Anal.* **37**, 89–105 (2014).
2. D Abov, K., Foi, A., Katkovnik, V. & Egiazarian, K. Image denoising with block-matching and 3D filtering. in *Proceedings of SPIE - The International Society for Optical Engineering* 354–365 (2006).
3. Liu, X. *et al.* Non-line-of-sight reconstruction with signal–object collaborative regularization. *Light Sci. Appl.* **10**, 198 (2021).

List of changes

1. We have revised some typos. **(Lines 63, 179 and 184)**
2. We have weakened the statement of the application range of the proposed method. **(Line 161)**
3. We have included the PF-BP reconstructions for comparisons in the revised Figure 7. **(Lines 297 and 410, Figure 7 and Table 1)**
4. We have added a sentence that includes three previous works that may help to understand the regularization terms introduced. **(Line 538)**
5. We have revised all the figures in the main manuscript so that they meet the publication standards. We have also updated the corresponding statements that were related to them.